# A Competitive Algorithm for Agnostic Active Learning

**Eric Price**
Department of Computer Science
University of Texas at Austin
ecprice@cs.utexas.edu

**Yihan Zhou**
Department of Computer Science
University of Texas at Austin
joeyzhou@cs.utexas.edu

## Abstract

For some hypothesis classes and input distributions, *active* agnostic learning needs exponentially fewer samples than passive learning; for other classes and distributions, it offers little to no improvement. The most popular algorithms for agnostic active learning express their performance in terms of a parameter called the disagreement coefficient, but it is known that these algorithms are inefficient on some inputs.

We take a different approach to agnostic active learning, getting an algorithm that is *competitive* with the optimal algorithm for any binary hypothesis class $H$ and distribution $\mathcal{D}_X$ over $X$. In particular, if any algorithm can use $m^*$ queries to get $O(\eta)$ error, then our algorithm uses $O(m^* \log |H|)$ queries to get $O(\eta)$ error. Our algorithm lies in the vein of the splitting-based approach of Dasgupta [2004], which gets a similar result for the realizable ($\eta = 0$) setting.

We also show that it is NP-hard to do better than our algorithm's $O(\log |H|)$ overhead in general.

## 1   Introduction

Active learning is motivated by settings where unlabeled data is cheap but labeling it is expensive. By carefully choosing which points to label, one can often achieve significant reductions in label complexity [Cohn et al., 1994]. A canonical example with exponential improvement is one-dimensional threshold functions $h_\tau(x) := 1_{x \geq \tau}$: in the noiseless setting, an active learner can use binary search to find an $\varepsilon$-approximation solution in $O\left(\log \frac{1}{\varepsilon}\right)$ queries, while a passive learner needs $\Theta\left(\frac{1}{\varepsilon}\right)$ samples [Cohn et al., 1994, Dasgupta, 2005, Nowak, 2011].

In this paper we are concerned with agnostic binary classification. We are given a hypothesis class $H$ of binary hypotheses $h : \mathcal{X} \to \{0, 1\}$ such that some $h^* \in H$ has $\mathrm{err}(h^*) \leq \eta$, where the error

$$\mathrm{err}(h) := \Pr_{(x,y) \sim \mathcal{D}}[h(x) \neq y]$$

is measured with respect to an unknown distribution $\mathcal{D}$ over $\mathcal{X} \times \{0, 1\}$. In our *active* setting, we also know the marginal distribution $\mathcal{D}_X$ of $x$, and can query any point $x$ of our choosing to receive a sample $y \sim (Y \mid X = x)$ for $(X, Y) \sim \mathcal{D}$. The goal is to output some $\widehat{h}$ with $\mathrm{err}(\widehat{h}) \leq \eta + \varepsilon$, using as few queries as possible.

The first interesting results for agnostic active learning were shown by Balcan et al. [2006], who gave an algorithm called Agnostic Active ($A^2$) that gets logarithmic dependence on $\varepsilon$ in some natural settings: it needs $\widetilde{O}\left(\log \frac{1}{\varepsilon}\right)$ samples for the 1d linear threshold setting (binary search), as long as as $\varepsilon > 16\eta$, and $\widetilde{O}\left(d^2 \log \frac{1}{\varepsilon}\right)$ samples for $d$-dimensional linear thresholds when $\mathcal{D}_X$ is the uniform sphere and $\varepsilon > \sqrt{d}\eta$. This stands in contrast to the polynomial dependence on $\varepsilon$ necessary in the passive setting. The bound's requirement that $\varepsilon \gtrsim \eta$ is quite natural given a lower bound of $\Omega\left(d\frac{\eta^2}{\varepsilon^2}\right)$

37th Conference on Neural Information Processing Systems (NeurIPS 2023).

due to [Kääriäinen, 2006, Beygelzimer et al., 2009], where $d$ is the VC dimension. Subsequent works have given new algorithms [Dasgupta et al., 2007, Beygelzimer et al., 2010] and new analyses [Hanneke, 2007a] to get bounds for more general problems, parameterized by the "disagreement coefficient" of the problem. But while these can give better bounds in specific cases, they do not give a good competitive ratio to the optimum algorithm: see (Hanneke [2014], Section 8.2.5) for a realizable example where $O\left(\log \frac{1}{\varepsilon}\right)$ queries are possible, but disagreement-coefficient based bounds lead to $\Omega\left(\frac{1}{\varepsilon}\right)$ queries.

By contrast, in the *realizable, identifiable* setting ($\eta = \varepsilon = 0$), a simple greedy algorithm *is* competitive with the optimal algorithm. In particular, Dasgupta [2004] shows that if any algorithm can identify the true hypothesis in $m$ queries, then the greedy algorithm that repeatedly queries the point that splits the most hypotheses will identify the true hypothesis in $O(m \log |H|)$ queries. This extra factor of $\log |H|$ is computationally necessary: as we will show in Theorem 1.2, avoiding it is NP-hard in general. This approach can extend [Dasgupta, 2005] to the PAC setting (so $\varepsilon > 0$, but still $\eta = 0$), showing that if any algorithm gets error $\varepsilon$ in $m^*$ queries, then this algorithm gets error $8\varepsilon$ in roughly $\widetilde{O}(m^* \cdot \log |H|)$ queries (but see the discussion after Theorem 8.2 of Hanneke [2014], which points out that one of the logarithmic factors is in an uncontrolled parameter $\tau$, and states that "Resolving the issue of this extra factor of $\log \frac{1}{\tau}$ remains an important open problem in the theory of active learning.").

The natural question is: can we find an agnostic active learning algorithm that is competitive with the optimal one in the agnostic setting?

**Our Results.**    Our main result is just such a competitive bound. We say an active agnostic learning algorithm $\mathcal{A}$ solves an instance $(H, \mathcal{D}_X, \eta, \varepsilon, \delta)$ with $m$ measurements if, for every distribution $\mathcal{D}$ with marginal $\mathcal{D}_X$ and for which some $h^* \in H$ has $\mathrm{err}(h^*) \leq \eta$, with probability $1 - \delta$, $\mathcal{A}$ uses at most $m$ queries and outputs $\widehat{h} \in H$ with $\mathrm{err}\left(\widehat{h}\right) \leq \eta + \varepsilon$. Let $m^*(H, \mathcal{D}_X, \eta, \varepsilon, \delta)$ be the optimal number of queries for this problem, i.e., the smallest $m$ for which any $\mathcal{A}$ can solve $(H, \mathcal{D}_X, \eta, \varepsilon, \delta)$.

Define $N(H, \mathcal{D}_X, \alpha)$ to be the size of the smallest $\alpha$-cover over $H$, i.e., the smallest set $S \subseteq H$ such that for every $h \in H$ there exists $h' \in S$ with $\mathrm{Pr}_{x \sim \mathcal{D}_X}[h(x) \neq h'(x)] \leq \alpha$. When the context is clear, we drop the parameters and simply use $N$. Of course, $N$ is at most $|H|$.

**Theorem 1.1** (Competitive Bound). *There exist some constants $c_1, c_2$ and $c_3$ such that for any instance $(H, \mathcal{D}_X, \eta, \varepsilon, \delta)$ with $\varepsilon \geq c_1 \eta$, Algorithm 1 solves the instance with sample complexity*

$$m(H, \mathcal{D}_X, \eta, \varepsilon, \delta) \lesssim \left( m^* \left( H, \mathcal{D}_X, c_2 \eta, c_3 \varepsilon, \frac{99}{100} \right) + \log \frac{1}{\delta} \right) \cdot \log \frac{N(H, \mathcal{D}_X, \eta)}{\delta}$$

*and polynomial time.*

Even the case of $\eta = 0$ is interesting, given the discussion in [Hanneke, 2014] of the gap in [Dasgupta, 2005]'s bound, but the main contribution is the ability to handle the agnostic setting of $\eta > 0$. The requirement that $\varepsilon \geq O(\eta)$ is in line with prior work [Balcan et al., 2006, Dasgupta, 2005]. Up to constants in $\eta$ and $\varepsilon$, Theorem 1.1 shows that our algorithm is within a $\log N \leq \log |H|$ factor of the optimal query complexity.

We show that it NP-hard to avoid this $\log N$ factor, even in the realizable ($\eta = \varepsilon = \delta = 0$) case:

**Theorem 1.2** (Lower Bound). *It is NP-hard to find a query strategy for every agnostic active learning instance within an $c \log |H|$ for some constant $c > 0$ factor of the optimal sample complexity.*

This is a relatively simple reduction from the hardness of approximating SETCOVER [Dinur and Steurer, 2014]. The lower bound instance has $\eta = \varepsilon = \delta = 0$, although these can be relaxed to being small polynomials (e.g., $\varepsilon = \eta = \frac{1}{3|X|}$ and $\delta = \frac{1}{3|H|}$).

**Extension.**    We give an improved bound for our algorithm in the case of noisy binary search (i.e., $H$ consists of 1d threshold functions). When $\eta = \Theta(\varepsilon)$, $N(H, \mathcal{D}_X, \varepsilon) = \Theta(\frac{1}{\varepsilon})$ and $m^*(\eta, \varepsilon, .99) = O(\log \frac{1}{\varepsilon})$. Thus Theorem 1.1 immediately gives a bound of $O(\log^2 \frac{1}{\varepsilon \delta})$, which is nontrivial but not ideal. (For $\eta \ll \varepsilon$, the same bound holds since the problem is strictly easier when $\eta$ is smaller.) However, the bound in Theorem 1.1 is quite loose in this setting, and we can instead give a bound of

$$O\left( \log \frac{1}{\varepsilon \delta} \log \frac{\log \frac{1}{\varepsilon}}{\delta} \right)$$

for the same algorithm, Algorithm 1. This matches the bound given by disagreement coefficient based algorithms for constant $\delta$. The proof of this improved dependence comes from bounding a new parameter measuring the complexity of an $H, \mathcal{D}_x$ pair; this parameter is always at least $\Omega(\frac{1}{m^*})$ but may be much larger (and is constant for 1d threshold functions). See Theorem 2.3 for details.

## 1.1 Related Work

Active learning is a widely studied topic, taking many forms beyond the directly related work on agnostic active learning discussed above [Settles, 2009]. Our algorithm can be viewed as similar to "uncertainty sampling" [Lewis, 1995, Lewis and Catlett, 1994], a popular empirical approach to active learning, though we need some modifications to tolerate adversarial noise.

One problem related to the one studied in this paper is noisy binary search, which corresponds to active learning of 1d thresholds. This has been extensively studied in the setting of *i.i.d.* noise [Burnashev and Zigangirov, 1974, Ben-Or and Hassidim, 2008, Dereniowski et al., 2021] as well as monotonic queries [Karp and Kleinberg, 2007]. Some work in this vein has extended beyond binary search to (essentially) active binary classification [Nowak, 2008, 2011]. These algorithms are all fairly similar to ours, in that they do multiplicative weights/Bayesian updates, but they query the *single* maximally informative point. This is fine in the i.i.d. noise setting, but in an agnostic setting the adversary can corrupt that query. For this reason, our algorithm needs to find a *set* of high-information points to query.

Another related problems is decision tree learning. The realizable, noiseless case $\eta = \varepsilon = 0$ of our problem can be reduced to learning a binary decision tree with minimal depth. Hegedűs [1995] studied this problem and gave basically the same upper and lower bound as in Dasgupta [2005]. Kosaraju et al. [2002] studied a split tree problem, which is a generalization of binary decision tree learning, and also gave similar bounds. Azad et al. [2022] is a monograph focusing on decision tree learning, in which many variations are studied, including learning with noise. However, this line of work usually allows different forms of queries so their results are not directly comparable from results in the active learning literature.

For much more work on the agnostic active binary classification problem, see Hanneke [2014] and references therein. Many of these papers give bounds in terms of the disagreement coefficient, but sometimes in terms of other parameters. For example, Katz-Samuels et al. [2021] has a query bound that is always competitive with the disagreement coefficient-based methods, and sometimes much better; still, it is not competitive with the optimum in all cases.

In terms of the lower bound, it is shown in Laurent and Rivest [1976] that the problem is NP-complete, in the realizable and noiseless setting. To the best of our knowledge, our Theorem 1.2 showing hardness of approximation to within a $O(\log |H|)$ factor is new.

**Minimax sample complexity bounds.**  Hanneke and Yang [2015] and Hanneke [2007b] have also given "minimax" sample complexity bounds for their algorithms, also getting a sample complexity within $O(\log |H|)$ of optimal. However, these results are optimal with respect to the sample complexity for the worst-case distribution over $y$ *and* $x$. But the unlabeled data $x$ is given as input. So one should hope for a bound with respect to optimal for the *actual* $x$ and only worst-case over $y$; this is our bound.

We give the following example to illustrate that our bound, and indeed our algorithm, can be much better.

**Example 1.3.** *Define a hypothesis class of $N$ hypotheses $h_1, \cdots, h_N$, and $\log N + N$ data points $x_1, \cdots, x_{\log N + N}$. For each hypothesis $h_j$, the labels of the first $N$ points express $j$ in unary and the labels of the last $\log N$ points express $j$ in binary. We set $\eta = \varepsilon = 0$ and consider the realizable case.*

In the above example, the binary region is far more informative than the unary region, but disagreement coefficient-based algorithms just note that every point has disagreement. Our algorithm will query the binary encoding region and take $O(\log N)$ queries. Disagreement coefficient based algorithms, including those in Hanneke and Yang [2015] and Hanneke [2007b], will rely on essentially uniform sampling for the first $\Omega(N/\log N)$ queries. These algorithms are "minimax" over $x$, in the sense that *if you didn't see any $x$ from the binary region*, you would need almost as many samples as they use. But you *do* see $x$ from the binary region, so the algorithm should make use of it to get exponential improvement.

**Future Work.** Our upper bound assumes full knowledge of $\mathcal{D}_X$ and the ability to query arbitrary points $x$. Often in active learning, the algorithm receives a large but not infinite set of unlabeled sample points $x$, and can only query the labels of those points. How well our results adapt to this setting we leave as an open question.

Similarly, our bound is polynomial in the number of hypotheses and the domain size. This is hard to avoid in full generality—if you don't evaluate most hypotheses on most data points, you might be missing the most informative points—but perhaps it can be avoided in structured examples.

## 2 Algorithm Overview

Our algorithm is based on a Bayesian/multiplicative weights type approach to the problem, and is along the lines of the splitting-based approach of Dasgupta [2004].

We maintain a set of weights $w(h)$ for each $h \in H$, starting at 1; these induce a distribution $\lambda(h) := \frac{w(h)}{\sum_h w(h)}$ which we can think of as our posterior over the "true" $h^*$.

**Realizable setting.** As initial intuition, consider the realizable case of $\eta = \varepsilon = 0$ where we want to find the true $h^*$. If $h^*$ really were drawn from our prior $\lambda$, and we query a point $x$, we will see a 1 with probability $\mathbb{E}_{h \sim \lambda} h(x)$. Then the most informative point to query is the one we are least confident in, i.e., the point $x^*$ maximizing

$$r(x) := \min \left\{ \mathbb{E}_{h \sim \lambda}[h(x)], 1 - \mathbb{E}_{h \sim \lambda}[h(x)] \right\}.$$

Suppose an algorithm queries $x_1, \ldots, x_m$ and receives the majority label under $h \sim \lambda$ each time. Then the fraction of $h \sim \lambda$ that agree with *all* the queries is at least $1 - \sum_{i=1}^m r(x_i) \geq 1 - mr(x^*)$. This suggests that, if $r(x^*) \ll \frac{1}{m}$, it will be hard to uniquely identify $h^*$. It is not hard to formalize this, showing that: if no single hypothesis has 75% probability under $\lambda$, and any algorithm exists with sample complexity $m$ and 90% success probability at finding $h^*$, we must have $r(x^*) \geq \frac{1}{10m}$.

This immediately gives an algorithm for the $\eta = \varepsilon = 0$ setting: query the point $x$ maximizing $r(x)$, set $w(h) = 0$ for all hypotheses $h$ that disagree, and repeat. As long as at least two hypotheses remain, the maximum probability will be $50\% < 90\%$ and each iteration will remove an $\Omega(\frac{1}{m})$ fraction of the remaining hypotheses; thus after $O(m \log H)$ rounds, only $h^*$ will remain. This is the basis for Dasgupta [2004].

**Handling noise: initial attempt.** There are two obvious problems with the above algorithm in the agnostic setting, where a (possibly adversarial) $\eta$ fraction of locations $x$ will not match $h^*$. First, a single error will cause the algorithm to forever reject the true hypothesis; and second, the algorithm makes deterministic queries, which means adversarial noise could be placed precisely on the locations queried to make the algorithm learn nothing.

To fix the first problem, we can adjust the algorithm to perform multiplicative weights: if in round $i$ we query a point $x_i$ and see $y_i$, we set

$$w_{i+1}(h) = \begin{cases} w_i(h) & \text{if } h(x_i) = y_i \\ e^{-\alpha} w_i(h) & \text{if } h(x_i) \neq y_i \end{cases}$$

for a small constant $\alpha = \frac{1}{5}$. To fix the second problem, we don't query the single $x^*$ of maximum $r(x^*)$, but instead choose $x$ according to distribution $q$ over many points $x$ with large $r(x)$.

To understand this algorithm, consider how $\log \lambda_i(h^*)$ evolves in expectation in each step. This increases if the query is correct, and decreases if it has an error. A correct query increases $\lambda_i$ in proportion to the fraction of $\lambda$ placed on hypotheses that get the query wrong, which is at least $r(x)$; and the probability of an error is at most $\eta \max_x \frac{q(x)}{\mathcal{D}_x(x)}$. If at iteration $i$ the algorithm uses query distribution $q$, some calculation gives that

$$\mathbb{E}_q \left[ \log \lambda_{i+1}(h^*) - \log \lambda_i(h^*) \right] \geq 0.9\alpha \left( \mathbb{E}_{x \sim q}[r(x)] - 2.3\eta \max_x \frac{q(x)}{\mathcal{D}_x(x)} \right). \tag{1}$$

|       | $\lambda(h)$     | Values $h(x)$ |
|-------|------------------|---------------|
| $h_1$ | 0.9              | 1111 1111     |
| $h_2$ | $0.1 - 10^{-6}$  | 1111 0000     |
| $h_3$ | $10^{-6}$        | 0000 1110     |
| $y$   |                  | 0000 1111     |

Figure 1: An example demonstrating that the weight of the true hypothesis can decrease if $\lambda$ is concentrated on the wrong ball. In this example, the true labels $y$ are closest to $h_3$. But if the prior $\lambda$ on hypotheses puts far more weight on $h_1$ and $h_2$, the algorithm will query uniformly over where $h_1$ and $h_2$ disagree: the second half of points. Over this query distribution, $h_1$ is more correct than $h_3$, so the weight of $h_3$ can actually *decrease* if $\lambda(h_1)$ is very large.

The algorithm can choose $q$ to maximize this bound on the potential gain. There's a tradeoff between concentrating the samples over the $x$ of largest $r(x)$, and spreading out the samples so the adversary can't raise the error probability too high. We show that if learning is possible by any algorithm (for a constant factor larger $\eta$), then there exists a $q$ for which this potential gain is significant.

**Lemma 2.1** (Connection to OPT). *Define* $\|h - h'\| = \Pr_{x \sim \mathcal{D}_x}[h(x) \neq h'(x)]$. *Let* $\lambda$ *be a distribution over* $H$ *such that no radius-*$(2\eta + \varepsilon)$ *ball* $B$ *centered on* $h \in H$ *has probability at least* 80%. *Let* $m^* = m^*\left(H, \mathcal{D}_X, \eta, \varepsilon, \frac{99}{100}\right)$. *Then there exists a query distribution* $q$ *over* $\mathcal{X}$ *with*

$$\mathbb{E}_{x \sim q}[r(x)] - \frac{1}{10}\eta \max_x \frac{q(x)}{\mathcal{D}_X(x)} \geq \frac{9}{100m^*}.$$

At a very high level, the proof is: imagine $h^* \sim \lambda$. If the algorithm only sees the majority label $y$ on every query it performs, then its output $\widehat{h}$ is independent of $h^*$ and cannot be valid for more than 80% of inputs by the ball assumption; hence a 99% successful algorithm must have a 19% chance of seeing a minority label. But for $m^*$ queries $x$ drawn with marginal distribution $q$, without noise the expected number of minority labels seen is $m^* \mathbb{E}[r(x)]$, so $\mathbb{E}[r(x)] \gtrsim 1/m^*$. With noise, the adversary can corrupt the minority labels in $h^*$ back toward the majority, leading to the given bound.

The query distribution optimizing (1) has a simple structure: take a threshold $\tau$ for $r(x)$, sample from $\mathcal{D}_x$ conditioned on $r(x) > \tau$, and possibly sample $x$ with $r(x) = \tau$ at a lower rate. This means the algorithm can efficiently find the optimal $q$.

Except for the caveat about $\lambda$ not already concentrating in a small ball, applying Lemma 2.1 combined with (1) shows that $\log \lambda(h^*)$ grows by $\Omega(\frac{1}{m^*})$ in expectation for each query. It starts out at $\log \lambda(h^*) = -\log H$, so after $O(m^* \log H)$ queries we would have $\lambda(h^*)$ being a large constant in expectation (and with high probability, by Freedman's inequality for concentration of martingales). Of course $\lambda(h^*)$ can't grow past 1, which features in this argument in that once $\lambda(h^*) > 80\%$, a small ball *will* have large probability and Lemma 2.1 no longer applies, but at that point we can just output any hypothesis in the heavy ball.

**Handling noise: the challenge.** There is one omission in the above argument that is surprisingly challenging to fix, and ends up requiring significant changes to the algorithm: if at an intermediate step $\lambda_i$ concentrates in the *wrong* small ball, the algorithm will not necessarily make progress. It is entirely possible that $\lambda_i$ concentrates in a small ball, even in the first iteration—perhaps 99% of the hypotheses in $H$ are close to each other. And if that happens, then we will have $r(x) \leq 0.01$ for most $x$, which could make the RHS of (1) negative for all $q$.

In fact, it seems like a reasonable Bayesian-inspired algorithm really must allow $\lambda(h^*)$ to decrease in some situations. Consider the setting of Figure 1. We have three hypotheses, $h_1, h_2,$ and $h_3$, and a prior $\lambda = (0.9, 0.099999, 10^{-6})$. Because $\lambda(h_3)$ is so tiny, the algorithm presumably should ignore $h_3$ and query essentially uniformly from the locations where $h_1$ and $h_2$ disagree. In this example, $h_3$ agrees with $h_1$ on all but an $\eta$ mass in those locations, so even if $h^* = h_3$, the query distribution can match $h_1$ perfectly and not $h_3$. Then $w(h_1)$ stays constant while $w(h_3)$ shrinks. $w(h_2)$ shrinks much faster, of course, but since the denominator is dominated by $w(h_1)$, $\lambda(h_3)$ will still shrink. However, despite $\lambda(h_3)$ shrinking, the algorithm is still making progress in this example: $\lambda(h_2)$ is shrinking fast, and once it becomes small relative to $\lambda(h_3)$ then the algorithm will start querying points to distinguish $h_3$ from $h_1$, at which point $\lambda(h_3)$ will start an inexorable rise.

Our solution is to "cap" the large density balls in $\lambda$, dividing their probability by two, when applying Lemma 2.1. Our algorithm maintains a set $S \subseteq H$ of the "high-density region," such that the capped

distribution:

$$\overline{\lambda}(h) := \begin{cases} \frac{1}{2}\lambda(h) & h \in S \\ \lambda(h) \cdot \frac{1 - \frac{1}{2}\Pr[h \in S]}{1 - \Pr[h \in S]} & h \notin S \end{cases}$$

has no large ball. Then Lemma 2.1 applies to $\overline{\lambda}$, giving the existence of a query distribution $q$ so that the corresponding $\overline{r}(x)$ is large. We then define the potential function

$$\phi_i(h^*) := \log \lambda_i(h^*) + \log \frac{\lambda_i(h^*)}{\sum_{h \notin S_i} \lambda_i(h)} \tag{2}$$

for $h^* \notin S_i$, and $\phi_i = 0$ for $h^* \in S_i$. We show that $\phi_i$ grows by $\Omega(\frac{1}{m^*})$ in expectation in each iteration. Thus, as in the example of Figure 1, either $\lambda(h^*)$ grows as a fraction of the whole distribution, or as a fraction of the "low-density" region.

If at any iteration we find that $\overline{\lambda}$ has some heavy ball $B(\mu, 2\eta + \varepsilon)$ so Lemma 2.1 would not apply, we add $B(\mu', 6\eta + 3\varepsilon)$ to $S$, where $B(\mu', 2\eta + \varepsilon)$ is the heaviest ball before capping. We show that this ensures that no small heavy ball exists in the capped distribution $\overline{\lambda}$. Expanding $S$ only increases the potential function, and then the lack of heavy ball implies the potential will continue to grow.

Thus the potential (2) starts at $-2\log|H|$, and grows by $\Omega(\frac{1}{m^*})$ in each iteration. After $O(m^* \log H)$ iterations, we will have $\phi_i \geq 0$ in expectation (and with high probability by Freedman's inequality). This is only possible if $h^* \in S$, which means that one of the centers $\mu$ of the balls added to $S$ is a valid answer.

In fact, with some careful analysis we can show that with $1 - \delta$ probability that one of the *first* $O(\log \frac{H}{\delta})$ balls added to $S$ is a valid answer. The algorithm can then check all the centers of these balls, using the following active agnostic learning algorithm:

**Theorem 2.2.** *Active agnostic learning can be solved for $\varepsilon = 3\eta$ with $O\left(|H| \log \frac{|H|}{\delta}\right)$ samples.*

*Proof.* The algorithm is the following. Take any pair $h, h'$ with $\|h - h'\| \geq 3\eta$. Sample $O\left(\log \frac{|H|}{\delta}\right)$ observations randomly from $(x \sim \mathcal{D}_x \mid h(x) \neq h'(x))$. One of $h, h'$ is wrong on at least half the queries; remove it from $H$ and repeat. At the end, return any remaining $h$.

To analyze this, let $h^* \in H$ be the hypothesis with error $\eta$. If $h^*$ is chosen in a round, the other $h'$ must have error at least $2\eta$. Therefore the chance we remove $h^*$ is at most $\delta/|H|$. In each round we remove a hypothesis, so there are at most $|H|$ rounds and at most $\delta$ probability of ever crossing off $h^*$. If we never cross off $h^*$, at the end we output some $h$ with $\|h - h^*\| \leq 3\eta$, which gives $\varepsilon = 3\eta$. $\square$

The linear dependence on $|H|$ makes the Theorem 2.2 algorithm quite bad in most circumstances, but the dependence *only* on $|H|$ makes it perfect for our second stage (where we have reduced to $O(\log|H|)$ candidate hypotheses).

Overall, this argument gives an $O\left(m^* \log \frac{|H|}{\delta} + \log \frac{|H|}{\delta} \log \frac{\log|H|}{\delta}\right)$ sample algorithm for agnostic active learning. One can simplify this bound by observing that the set of centers $C$ added by our algorithm form a packing, and must therefore all be distinguishable by the optimal algorithm, so $m^* \geq \log C$. This gives a bound of

$$O\left((m^* + \log \frac{1}{\delta}) \log \frac{|H|}{\delta}\right).$$

By starting with an $\eta$-net of size $N$, we can reduce $|H|$ to $N$ with a constant factor increase in $\eta$.

With some properly chosen constants $c_4$ and $c_5$, the entire algorithm is formally described in Algorithm 1.

**Remark 1:** As stated, the algorithm requires knowing $m^*$ to set the target sample complexity / number of rounds $k$. This restriction could be removed with the following idea. $m^*$ only enters the analysis through the fact that $O\left(\frac{1}{m^*}\right)$ is a lower bound on the expected increase of the potential

function in each iteration. However, the algorithm *knows* a bound on its expected increase in each round $i$; it is the value

$$\tau_i = \max_q \mathbb{E}_{x \sim q} \left[ \overline{r}_{i,S_i}(x) \right] - \frac{c_4}{20} \eta \max_x \frac{q(x)}{\mathcal{D}_X(x)}.$$

optimized in the algorithm. Therefore, we could use an adaptive termination criterion that stops at iteration $k$ if $\sum_{i=1}^{k} \tau_i \geq O(\log \frac{|H|}{\delta})$. This will guarantee that when terminating, the potential will be above 0 with high probability so our analysis holds.

**Remark 2:** The algorithm's running time is polynomial in $|H|$. This is in general not avoidable, since the input is a truth table for $H$. The bottleneck of the computation is the step where the algorithm checks if the heaviest ball has mass greater than 80%. This step could be accelerated by randomly sampling hypothesis and points to estimate and find heavy balls; this would improve the dependence to nearly linear in $|H|$. If the hypothesis class has some structure, like the binary search example, the algorithm can be implemented more efficiently.

---

**Algorithm 1** Competitive Algorithm for Active Agnostic Learning

---

Compute a $2\eta$ maximal packing $H'$
Let $w_0 = 1$ for every $h \in H'$.
$S_0 \leftarrow \emptyset$
$C \leftarrow \emptyset$
**for** $i = 1, \ldots, k = O\left( m^* \log \frac{|H'|}{\delta} \right)$ **do**

    Compute $\lambda_i(h) = \frac{w_{i-1}(h)}{\sum_{h \in H} w_{i-1}(h)}$ for every $h \in H$

    **if** there exists $c_4 \eta + c_5 \varepsilon$ ball with probability $> 80\%$ over $\overline{\lambda}_{i,S_{i-1}}$ **then**

        $S_i \leftarrow S_i \cup B\left(\mu', 3c_4\eta + 3c_5\varepsilon\right)$ where $B\left(\mu', c_4\eta + c_5\varepsilon\right)$ is the heaviest radius $c_4\eta + c_5\varepsilon$ ball over $\lambda_i$
        $C \leftarrow C \cup \{\mu'\}$

    **else**

        $S_i \leftarrow S_{i-1}$

    Compute $\overline{\lambda}_{i,S_i} = \begin{cases} \frac{1}{2} \lambda_i(h) & h \in S_i \\ \lambda_i(h) \cdot \frac{1 - \frac{1}{2} \Pr_{h \sim \lambda_i}[h \in S_i]}{1 - \Pr_{h \sim \lambda_i}[h \in S_i]} & h \notin S_i \end{cases}$

    Compute $\overline{r}_{i,S_i}(x) = \min\left\{ \mathbb{E}_{h \sim \overline{\lambda}_{i,S_i}}[h(x)], 1 - \mathbb{E}_{h \sim \overline{\lambda}_{i,S_i}}[h(x)] \right\}$ for every $x \in \mathcal{X}$

    Find a query distribution by solving

$$q^* = \max_q \mathbb{E}_{x \sim q}\left[ \overline{r}_{i,S_i}(x) \right] - \frac{c_4}{20} \eta \max_x \frac{q(x)}{\mathcal{D}_X(x)} \quad \text{subject to} \quad \mathbb{E}_{x \sim \mathcal{D}_X}[q(x)] \leq 3\eta \quad (3)$$

    Query $x \sim q^*$, getting label $y$

    Set $w_i(h) = \begin{cases} w_{i-1}(h) & \text{if } h(x) = y \\ e^{-\alpha} w_{i-1}(h) & \text{if } h(x) \neq y \end{cases}$ for every $h \in H'$

Find the best hypothesis $\hat{h}$ in $C$ using the stage two algorithm in Theorem 2.2
**return** $\hat{h}$

---

**Generalization for Better Bounds.** To get a better dependence for 1d threshold functions, we separate out the Lemma 2.1 bound on (1) from the analysis of the algorithm given a bound on (1). Then for particular instances like 1d threshold functions, we get a better bound on the algorithm by giving a larger bound on (1).

**Theorem 2.3.** *Suppose that $\mathcal{D}_x$ and $H$ are such that, for any distribution $\lambda$ over $H$ such that no radius-$(c_4\eta + c_5\varepsilon)$ ball has probability more than 80%, there exists a distribution $q$ over $X$ such that*

$$\mathbb{E}_{x \sim q}[r(x)] - \frac{c_4}{20} \eta \max_x \frac{q(x)}{\mathcal{D}_x(x)} \geq \beta$$

*for some $\beta > 0$. Then for $\varepsilon \geq c_1\eta$, $c_4 \geq 300$, $c_5 = \frac{1}{10}$ and $c_1 \geq 90c_4$, let $N = N(H, \mathcal{D}_x, \eta)$ be the size of an $\eta$-cover of $H$. Algorithm 1 solves $(\eta, \varepsilon, \delta)$ active agnostic learning with $O\left( \frac{1}{\beta} \log \frac{N}{\delta} + \log \frac{N}{\delta} \log \frac{\log N}{\delta} \right)$ samples.*

**Corollary 2.4.** *There exists a constant $c_1 > 1$ such that, for $1d$ threshold functions and $\varepsilon > c_1 \eta$, Algorithm 1 solves $(\eta, \varepsilon, \delta)$ active agnostic learning with $O\left(\log \frac{1}{\varepsilon \delta} \log \frac{\log \frac{1}{\varepsilon}}{\delta}\right)$ samples.*

*Proof.* Because the problem is only harder if $\eta$ is larger, we can raise $\eta$ to be $\eta = \varepsilon / C$, where $C > 1$ is a sufficiently large constant that Theorem 2.3 applies. Then $1d$ threshold functions have an $\eta$-cover of size $N = O(1/\varepsilon)$. To get the result by Theorem 2.3, it suffices to show $\beta = \Theta(1)$.

Each hypothesis is of the form $h(x) = 1_{x \geq \tau}$, and corresponds to a threshold $\tau$. So we can consider $\lambda$ to be a distribution over $\tau$.

Let $\lambda$ be any distribution for which no radius-$R$ with probability greater than $80\%$ ball exists, for $R = c_4 \eta + c_5 \varepsilon$. For any percent $p$ between 0 and 100, let $\tau_p$ denote the pth percentile of $\tau$ under $\lambda$ (i.e., the smallest $t$ such that $\Pr[\tau \leq t] \geq p/100$). By the ball assumption, $\tau_{10}$ and $\tau_{90}$ do not lie in the same radius-$R$ ball. Hence $\|h_{\tau_{10}} - h_{\tau_{90}}\| > R$, or

$$\Pr_x[\tau_{10} \leq x < \tau_{90}] > R.$$

We let $q$ denote $(\mathcal{D}_x \mid \tau_{10} \leq x < \tau_{90})$. Then for all $x \in \operatorname{supp}(q)$ we have $r(x) \geq 0.1$ and

$$\frac{q(x)}{D_x(x)} = \frac{1}{\Pr_{x \sim D_x}[x \in \operatorname{supp}(q)]} < \frac{1}{R}.$$

Therefore we can set

$$\beta = \mathbb{E}_{x \sim q}[r(x)] - \frac{c_4}{20} \eta \max_x \frac{q(x)}{D_x(x)} \geq 0.1 - \frac{c_4 \eta}{20(c_4 \eta + c_5 \varepsilon)} \gtrsim 1,$$

as needed. $\qquad \square$

## 3   Proof of Lemma 2.1

**Lemma 2.1** (Connection to OPT). *Define $\|h - h'\| = \Pr_{x \sim \mathcal{D}_x}[h(x) \neq h'(x)]$. Let $\lambda$ be a distribution over $H$ such that no radius-$(2\eta + \varepsilon)$ ball $B$ centered on $h \in H$ has probability at least $80\%$. Let $m^* = m^*\left(H, \mathcal{D}_X, \eta, \varepsilon, \frac{99}{100}\right)$. Then there exists a query distribution $q$ over $\mathcal{X}$ with*

$$\mathbb{E}_{x \sim q}[r(x)] - \frac{1}{10} \eta \max_x \frac{q(x)}{\mathcal{D}_X(x)} \geq \frac{9}{100 m^*}.$$

*Proof.* WLOG, we assume that $\Pr_{h \sim \lambda}[h(x) = 0] \geq \Pr_{h \sim \lambda}[h(x) = 1]$ for every $x \in \mathcal{X}$. This means $r(x) = \mathbb{E}_{h \sim \lambda}[h(x)]$. This can be achieved by flipping all $h(x)$ and observations $y$ for all $x$ not satisfying this property; such an operation doesn't affect the lemma statement.

We will consider an adversary defined by a function $g : X \to [0, 1]$. The adversary takes a hypothesis $h \in H$ and outputs a distribution over $y \in \{0, 1\}^X$ such that $0 \leq y(x) \leq h(x)$ always, and $\operatorname{err}(h) = \mathbb{E}_{x,y}[h(x) - y(x)] \leq \eta$ always. For a hypothesis $h$, the adversary sets $y(x) = 0$ for all $x$ with $h(x) = 0$, and $y(x) = 0$ independently with probability $g(x)$ if $h(x) = 1$—unless $\mathbb{E}_x[h(x)g(x)] > \eta$, in which case the adversary instead simply outputs $y = h$ to ensure the expected error is at most $\eta$ always.

We consider the agnostic learning instance where $x \sim \mathcal{D}_x$, $h \sim \lambda$, and $y$ is given by this adversary. Let $\mathcal{A}$ be an $(\eta, \varepsilon)$ algorithm which uses $m$ measurements and succeeds with $99\%$ probability. Then it must also succeed with $99\%$ probability over this distribution. For the algorithm to succeed on a sample $h$, its output $\widehat{h}$ must have $\|h - \widehat{h}\| \leq 2\eta + \varepsilon$. By the bounded ball assumption, for any choice of adversary, no fixed output succeeds with more than $80\%$ probability over $h \sim \lambda$.

Now, let $\mathcal{A}_0$ be the behavior of $\mathcal{A}$ if it observes $y = 0$ for all its queries, rather than the truth; $\mathcal{A}_0$ is independent of the input. $\mathcal{A}_0$ has some distribution over $m$ queries, outputs some distribution of answers $\widehat{h}$. Let $q(x) = \frac{1}{m} \Pr[\mathcal{A}_0 \text{ queries } x]$, so $q$ is a distribution over $\mathcal{X}$. Since $\mathcal{A}_0$ outputs a fixed distribution, by the bounded ball assumption, for $h \sim \lambda$ and arbitrary adversary function $g$,

$$\Pr_{h \sim \lambda}[\mathcal{A}_0 \text{ succeeds}] \leq 80\%.$$

But $\mathcal{A}$ behaves identically to $\mathcal{A}_0$ until it sees its first nonzero $y$. Thus,

$$99\% \leq \Pr[\mathcal{A} \text{ succeeds}] \leq \Pr[\mathcal{A}_0 \text{ succeeds}] + \Pr[\mathcal{A} \text{ sees a non-zero } y]$$

and so

$$\Pr[\mathcal{A} \text{ sees a non-zero } y] \geq 19\%.$$

Since $\mathcal{A}$ behaves like $\mathcal{A}_0$ until the first nonzero, we have

$$
\begin{aligned}
19\% &\leq \Pr[\mathcal{A} \text{ sees a non-zero } y] \\
&= \Pr[\mathcal{A}_0 \text{ makes a query } x \text{ with } y(x) = 1] \\
&\leq \mathbb{E}[\text{Number queries } x \text{ by } \mathcal{A}_0 \text{ with } y(x) = 1] \\
&= m \mathop{\mathbb{E}}_{h \sim \lambda} \mathop{\mathbb{E}}_{y} \mathop{\mathbb{E}}_{x \sim q} [y(x)].
\end{aligned}
$$

As an initial note, observe that $\mathbb{E}_{h,y}[y(x)] \leq \mathbb{E}_h[h(x)] = r(x)$ so

$$\mathop{\mathbb{E}}_{x \sim q}[r(x)] \geq \frac{0.19}{m}.$$

Thus the lemma statement holds for $\eta = 0$.

**Handling $\eta > 0$.** Consider the behavior when the adversary's function $g : X \to [0,1]$ satisfies $\mathbb{E}_{x \sim \mathcal{D}_x}[g(x)r(x)] \leq \eta/10$. We denote the class of all adversary satisfying this condition as $G$. We have that

$$\mathop{\mathbb{E}}_{h \sim \lambda}\left[\mathop{\mathbb{E}}_{x \sim \mathcal{D}_x}[g(x)h(x)]\right] = \mathop{\mathbb{E}}_{x \sim \mathcal{D}_x}[g(x)r(x)] \leq \eta/10.$$

Let $E_h$ denote the event that $\mathbb{E}_{x \sim \mathcal{D}_x}[g(x)h(x)] \leq \eta$, so $\Pr[\overline{E}_h] \leq 10\%$. Furthermore, the adversary is designed such that under $E_h$, $\mathbb{E}_y[y(x)] = h(x)(1 - g(x))$ for every $x$. Therefore:

$$
\begin{aligned}
0.19 &\leq \Pr[\mathcal{A}_0 \text{ makes a query } x \text{ with } y(x) = 1] \\
&\leq \Pr[\overline{E}_h] + \Pr[\mathcal{A}_0 \text{ makes a query } x \text{ with } y(x) = 1 \cap E_h] \\
&\leq 0.1 + \mathbb{E}[\text{Number queries } x \text{ by } \mathcal{A}_0 \text{ with } y(x) = 1 \text{ and } E_h] \\
&= 0.1 + m \mathop{\mathbb{E}}_h\left[\mathbb{1}_{E_h} \mathop{\mathbb{E}}_{x \sim q}[\mathop{\mathbb{E}}_y y(x)]\right] \\
&= 0.1 + m \mathop{\mathbb{E}}_h\left[\mathbb{1}_{E_h} \mathop{\mathbb{E}}_{x \sim q}[h(x)(1 - g(x))]\right] \\
&\leq 0.1 + m \mathop{\mathbb{E}}_{x \sim q}[\mathop{\mathbb{E}}_h[h(x)](1 - g(x))] \\
&= 0.1 + m \mathop{\mathbb{E}}_{x \sim q}[r(x)(1 - g(x))].
\end{aligned}
$$

Thus

$$\max_q \min_{g \in G} \mathop{\mathbb{E}}_{x \sim q}[r(x)(1 - g(x))] \geq \frac{9}{100m} \tag{4}$$

over all distributions $q$ and functions $g : X \to [0,1]$ satisfying $\mathbb{E}_{x \sim \mathcal{D}_x}[g(x)r(x)] \leq \eta/10$. We now try to understand the structure of the $q, g$ optimizing the LHS of (4).

Let $g^*$ denote an optimizer of the objective. First, we show that the constraint is tight, i.e., $\mathbb{E}_{x \sim \mathcal{D}_x}[g^*(x)r(x)] = \eta/10$. Since increasing $g$ improves the constraint, the only way this could not happen is if the maximum possible function, $g(x) = 1$ for all $x$, lies in $G$. But for this function, the LHS of (4) would be 0, which is a contradiction; hence we know increasing $g$ to improve the objective at some point hits the constraint, and hence $\mathbb{E}_{x \sim \mathcal{D}_x}[g^*(x)r(x)] = \eta/10$.

For any $q$, define $\tau_q \geq 0$ to be the minimum threshold such that

$$\mathop{\mathbb{E}}_{x \sim \mathcal{D}_x}\left[r(x) \cdot \mathbb{1}_{\frac{q(x)}{\mathcal{D}_X(x)} > \tau_q}\right] < \eta/10.$$

and define $g_q$ by

$$
g_q(x) := \begin{cases} 1 & \frac{q(x)}{\mathcal{D}_X(x)} > \tau_q \\ \alpha & \frac{q(x)}{\mathcal{D}_X(x)} = \tau_q \\ 0 & \frac{q(x)}{\mathcal{D}_X(x)} < \tau_q \end{cases}
$$

where $\alpha \in [0,1]$ is chosen such that $\mathbb{E}_{x \sim \mathcal{D}_x}[r(x)g_q(x)] = \eta/10$; such an $\alpha$ always exists by the choice of $\tau_q$.

For any $q$, we claim that the optimal $g^*$ in the LHS of (4) is $g_q$. It needs to maximize

$$\mathbb{E}_{x \sim \mathcal{D}_X} \left[ \frac{q(x)}{\mathcal{D}_X(x)} r(x) g(x) \right]$$

subject to a constraint on $\mathbb{E}_{x \sim \mathcal{D}_X}[r(x)g(x)]$; therefore moving mass to points of larger $\frac{q(x)}{\mathcal{D}_X(x)}$ is always an improvement, and $g_q$ is optimal.

We now claim that the $q$ maximizing (4) has $\max \frac{q(x)}{\mathcal{D}_X(x)} = \tau_q$. If not, some $x'$ has $\frac{q(x')}{\mathcal{D}_X(x')} > \tau_q$. Then $g_q(x') = 1$, so the $x'$ entry contributes nothing to $\mathbb{E}_{x \sim q}[r(x)(1 - g_q(x))]$; thus decreasing $q(x)$ halfway towards $\tau_q$ (which wouldn't change $g_q$), and adding the savings uniformly across all $q(x)$ (which also doesn't change $g_q$) would increase the objective.

So there exists a $q$ satisfying (4) for which $\Pr\left[\frac{q(x)}{\mathcal{D}_X(x)} > \tau_q\right] = 0$, and therefore the set $T = \left\{ x \mid \frac{q(x)}{\mathcal{D}_X(x)} = \tau_q \right\}$ satisfies $\mathbb{E}_{\mathcal{D}_X}[r(x)\mathbb{1}_{x \in T}] \geq \eta/10$ and a $g_q$ minimizing (4) is

$$g_q(x) = \frac{\eta}{10} \frac{\mathbb{1}_{x \in T}}{\mathbb{E}_{\mathcal{D}_X}[r(x)\mathbb{1}_{x \in T}]}.$$

Therefore

$$\mathbb{E}_{x \sim q}[r(x)g_q(x)] = \mathbb{E}_{x \sim \mathcal{D}_X} \left[ \frac{q(x)}{\mathcal{D}_X(x)} r(x) \frac{\eta}{10} \frac{\mathbb{1}_{x \in T}}{\mathbb{E}_{\mathcal{D}_X}[r(x)\mathbb{1}_{x \in T}]} \right]$$

$$\leq \frac{\eta}{10} \max_x \frac{q(x)}{\mathcal{D}_X(x)}$$

and so by (4),

$$\mathbb{E}_{x \sim q}[r(x)] - \frac{\eta}{10} \max_x \frac{q(x)}{\mathcal{D}_X(x)} \geq \frac{9}{100m}$$

as desired. $\qquad\square$

## 4 Conclusion

We have given an algorithm that solves agnostic active learning with (for constant $\delta$) at most an $O(\log|H|)$ factor more queries than the optimal algorithm. It is NP-hard to improve upon this $O(\log|H|)$ factor in general, but for specific cases it can be avoided. We have shown that 1d threshold functions, i.e. binary search with adversarial noise, is one such example where our algorithm matches the performance of disagreement coefficient-based methods and is within a $\log\log\frac{1}{\varepsilon}$ factor of optimal.

## 5 Acknowledgments

Yihan Zhou and Eric Price were supported by NSF awards CCF-2008868, CCF-1751040 (CAREER), and the NSF AI Institute for Foundations of Machine Learning (IFML).

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

# A  Query Complexity Upper Bound

In this section we present the whole proof of the query complexity upper bound of Algorithm 1, as stated in Theorem 1.1.

## A.1  Notation

We remind the readers about some definitions first. Remember that $w_i(h)$ denote the weight of hypothesis $h$ in iteration $i$ and $\lambda_{i,S}(h) = \frac{w_i(h)}{\sum_{h' \in S} w_i(h')}$ for some $S \subseteq H$ denote the proportion of $h$ in $S$. We view $\lambda_{i,S}$ as a distribution of hypotheses in $S$ so for $h \notin S$, $\lambda_{i,S}(h) = 0$. For a set $S \subseteq H$ of hypotheses, we define $w_i(S) := \sum_{h \in S} w(h)$ and $\lambda_i(h) = \lambda_{i,H}(h)$.

Define $r_{\lambda,h^*}(x) := \Pr_{h \sim \lambda}[h(x) \neq h^*(x)]$, and $r_\lambda(x) = \min_{y \in \{0,1\}} \Pr_{h \sim \lambda}[h(x) \neq y]$, so $r_\lambda(x) = \min(r_{\lambda,h^*}(x), 1 - r_{\lambda,h^*}(x))$.

Define

$$\overline{\lambda}_{i,S}(h) := \frac{1}{2}\lambda_i(h) + \frac{1}{2}\lambda_{i,H\setminus S}(h) = \begin{cases} \frac{1}{2}\lambda_i(h) & h \in S \\ \lambda_i(h) \cdot \frac{1 - \frac{1}{2}\Pr_{h\sim\lambda_i}[h \in S]}{1 - \Pr_{h\sim\lambda_i}[h \in S]} & h \notin S \end{cases} \tag{5}$$

as the "capped" distribution in iteration $i$.

Finally, for notational convenience define $r_{i,S} := r_{\lambda_{i,S}}$, $r_{i,S,h} := r_{\lambda_{i,S},h}$ and $\overline{r}_{i,S} := r_{\overline{\lambda}_{i,S}}$.

The main focus of our proof would be analyzing the potential function

$$\phi_i(h^*) = \begin{cases} \log \lambda_i(h^*) + \log \lambda_{i,H\setminus S_i}(h^*) & h^* \notin S_i \\ 0 & h^* \in S_i, \end{cases}$$

where $h^*$ is the best hypothesis in $H$. We would like to show that $\phi_{i+1}(h^*) - \phi_i(h^*)$ is growing at a proper rate in each iteration. We pick $S_i$ to be an expanding series of sets, i.e., $S_i \subseteq S_{i+1}$ for any $i \geq 1$. However, the change of the "capped" set $S_i$ makes this task challenging. Therefore, we instead analyze the following quantity defined as

$$\Delta_i(h^*) := \begin{cases} \log \frac{\lambda_{i+1}(h^*)}{\lambda_i(h^*)} + \log \frac{\lambda_{i+1,H\setminus S_i}(h^*)}{\lambda_{i,H\setminus S_i}(h^*)} & h^* \notin S_i \\ 0 & h^* \in S_i, \end{cases}$$

and $\phi_{i+1}(h^*) - \phi_i(h^*) = \Delta_i(h^*) + \log \frac{\lambda_{i+1,H\setminus S_{i+1}}(h^*)}{\lambda_{i+1,H\setminus S_i}(h^*)}$ if $h^* \notin S_{i+1}$. Further, we define $\psi_k(h^*) := \sum_{i<k} \Delta_i(h^*)$ so by definition $\phi_k(h^*) = \phi_0(h^*) + \psi_k(h^*) + \sum_{i<k} \log \frac{\lambda_{i+1,H\setminus S_{i+1}}(h^*)}{\lambda_{i+1,H\setminus S_i}(h^*)}$ if $h^* \notin S_{i+1}$. In the following text, we will drop the parameter $h^*$ when the context is clear and just use $\phi_i$, $\Delta_i$ and $\psi_i$ instead.

## A.2  Potential Growth

We will lower bound the conditional per iteration potential increase by first introducing a lemma that relates the potential change to the optimization problem (3).

**Lemma A.1.** *Assume that* $\mathrm{err}(h^*) \leq \eta$, *then for any set $S$ of hypotheses containing $h^*$ and query distribution $q$, we have*

$$\mathbb{E}\left[\log \frac{\lambda_{i+1,S}(h^*)}{\lambda_{i,S}(h^*)} \middle| \mathcal{F}_i\right] \geq 0.9\alpha \left(\mathbb{E}_{x\sim q}[r_{i,S,h}(x)] - 2.3\eta \max_x \frac{q(x)}{D_X(x)}\right)$$

*for $\alpha \leq 0.2$. Moreover,*

$$\mathbb{E}\left[\max\left\{0, \log \frac{\lambda_{i+1,S}(h^*)}{\lambda_{i,S}(h^*)}\right\} \middle| \mathcal{F}_i\right] \leq \alpha \mathbb{E}_{x\sim q}[r_{i,S,h^*}(x)].$$

*Proof.* For notational convenience, define $\widetilde{r}(x) := r_{i,S,h^*}(x)$.

Observe that

$$\frac{\lambda_{i,S}(h^*)}{\lambda_{i+1,S}(h^*)} = \frac{w_i(h^*)}{w_{i+1}(h^*)} \frac{\sum_{h\in S} w_{i+1,S}(h)}{\sum_{h\in S} w_{i,S}(h)} = \frac{w_i(h^*)}{w_{i+1}(h^*)} \mathbb{E}_{h\sim\lambda_{i,S}}\left[\frac{w_{i+1,S}(h)}{w_{i,S}(h)}\right].$$

Let $p(x) = \Pr_{y \sim (Y|X)}[y \neq h^*(x)]$ denote the probability of error if we query $x$, so

$$\mathbb{E}_{x \sim \mathcal{D}_X}[p(x)] \leq \eta.$$

Suppose we query a point $x$ and do not get an error. Then the hypotheses that disagree with $h^*$ are downweighted by an $e^{-\alpha}$ factor, so

$$\frac{\lambda_{i,S}(h^*)}{\lambda_{i+1,S}(h^*)} = \mathbb{E}_{h \sim \lambda_{i,S}}[1 + (e^{-\alpha} - 1)1_{h(x) \neq h^*(x)}] = 1 - (1 - e^{-\alpha})\widetilde{r}(x).$$

On the other hand, if we do get an error then the disagreeing hypotheses are effectively upweighted by $e^{\alpha}$:

$$\frac{\lambda_{i,S}(h^*)}{\lambda_{i+1,S}(h^*)} = 1 + (e^{\alpha} - 1)\widetilde{r}(x).$$

Therefore

$$
\begin{aligned}
\mathbb{E}_{y|x}&\left[\log \frac{\lambda_{i+1,S}(h^*)}{\lambda_{i,S}(h^*)}\middle| \mathcal{F}_i\right] \\
&= -(1 - p(x))\log\left(1 - (1 - e^{-\alpha})\widetilde{r}(x)\right) - p(x)\log\left(1 + (e^{\alpha} - 1)\widetilde{r}(x)\right) \quad (6)\\
&\geq (1 - p(x))(1 - e^{-\alpha})\widetilde{r}(x) - p(x)(e^{\alpha} - 1)\widetilde{r}(x) \\
&= (1 - e^{-\alpha})\widetilde{r}(x) - p(x)\widetilde{r}(x)(e^{\alpha} - e^{-\alpha}).
\end{aligned}
$$

Using that $\widetilde{r}(x) \leq 1$, we have

$$
\begin{aligned}
\mathbb{E}\left[\log \frac{\lambda_{i+1,S}(h^*)}{\lambda_{i,S}(h^*)}\middle| \mathcal{F}_i\right] &\geq (1 - e^{-\alpha})\mathbb{E}_{x \sim q}[\widetilde{r}(x)] - (e^{\alpha} - e^{-\alpha})\mathbb{E}_{x \sim q}[p(x)] \\
&\geq 0.9\alpha\,\mathbb{E}_{x \sim q}[\widetilde{r}(x) - 2.3p(x)],
\end{aligned}
$$

where the last step uses $\alpha \leq 0.2$. Finally,

$$\mathbb{E}_{x \sim q}[p(x)] = \mathbb{E}_{x \sim \mathcal{D}_X}\left[p(x)\frac{q(x)}{\mathcal{D}_X(x)}\right] \leq \eta \max_x \frac{q(x)}{\mathcal{D}_X(x)}.$$

This proves the first desired result. For the second, note that if we query $x$, then conditioned on $\mathcal{F}_i$

$$\max\left\{0, \log \frac{\lambda_{i+1,S}(h^*)}{\lambda_{i,S}(h^*)}\right\} = \begin{cases} 0 & \text{with probability } p(x), \\ \log(1 + (1 - e^{-\alpha})\widetilde{r}(x)) & \text{otherwise.} \end{cases}$$

Since $\log(1 + (1 - e^{-\alpha})\widetilde{r}(x)) \leq (1 - e^{-\alpha})\widetilde{r}(x) \leq \alpha\widetilde{r}(x)$, taking the expectation over $x$ gives the result. $\qquad\square$

The above lemma, combined with Lemma 2.1, proves the potential will grow at desired rate at each iteration. But remember that Lemma 2.1 requires the condition that no ball has probability greater than $80\%$, so we need to check this condition is satisfied. The following lemma shows that if we cap the set $S_i$, then the probability is not concentrated on any small balls.

**Lemma A.2.** *In Algorithm 1, for every iteration $i$, $S_i$ is such that no radius $c_4\eta + c_5\varepsilon$ ball has more than $80\%$ probability under $\overline{\lambda}_{i,S_i}$.*

*Proof.* If $S_i = S_{i-1}$, then by the construction of $S_i$, there are no radius $c_4\eta + c_5\varepsilon$ balls have probability greater than $80\%$ under $\overline{\lambda}_{i,S_{i-1}} = \overline{\lambda}_{i,S_i}$. Otherwise, we have $S_{i-1} \neq S_i$ and a ball $B(\mu, 3c_4\eta + 3c_5\varepsilon)$ is added to $S_i$ in this iteration. We first prove a useful claim below.

**Claim A.3.** *If a ball $B' = (\mu, 3c_4\eta + 3c_5\varepsilon)$ is added to $S_i$ at some iteration $i$, $\lambda_i(B(\mu, c_4\eta + c_5\varepsilon)) \geq 0.6$.*

*Proof.* If $B'$ is added to $S_i$ at the iteration $i$, then there exists some ball $D$ with radius $c_4\eta + c_5\varepsilon$ such that $\overline{\lambda}_{i,S_{i-1}}(D) \geq 0.8$. If a set of hypotheses gains probability after capping, the gained probability comes from the reduced probability of other hypotheses not in this set. Therefore, the gained probability of any set is upper bounded by half of the probability of the complement of that set before capping. This means $\lambda_i(D) \geq 0.6$ because otherwise after capping $\overline{\lambda}_{i,S_{i-1}}(D) < 0.8$, which is a contradiction. As a result, $\lambda_i(B(\mu, c_4\eta + c_5\varepsilon)) \geq \lambda_i(D) \geq 0.6$. $\qquad\square$

By Claim A.3, the probability of $B(\mu, c_4\eta + c_5\varepsilon)$ is at least $0.6$ over the uncapped distribution $\lambda_i$. So any ball not intersecting $B(\mu, c_4\eta + c_5\varepsilon)$ has probability at most $0.4$ before capping. After capping these balls will have probability no more than $0.7$. At the same time, any ball intersects $B(\mu, c_4\eta + c_5\varepsilon)$ would be completely inside $B(\mu, 3c_4\eta + 3c_5\varepsilon)$ so its probability would be at most $0.5$ after capping. $\qquad\square$

Now we are ready to apply Lemma A.1 and Lemma 2.1 except one caution. Remember that in the beginning of the algorithm, we compute a $2\eta$-packing $H' \subseteq H$ of the instance. From the well-known relationship between packing and covering (for example, see Vershynin [2018, Lemma 4.2.8]), we have $|H'| \leq N(H, \eta)$. Every hypothesis in $H$ is within $2\eta$ to some hypothesis in $H'$, so there exists a hypothesis in $H'$ with error less than $3\eta$. This means that the best hypothesis $h^* \in H'$ has error $3\eta$ instead of $\eta$. The following lemma serves as the cornerstone of the proof of the query complexity upper bound, which states that the potential grows at rate $\Omega\left(\frac{1}{m^*}\right)$ in each iteration.

**Lemma A.4.** *Given $c_4 \geq 300$ and $\mathrm{err}(h^*) \leq 3\eta$, there exists a sampling distribution $q$ such that*

$$\mathbb{E}[\Delta_i | \mathcal{F}_i] \geq \mathbb{E}\left[\Delta_i | \mathcal{F}_i\right] - 2\alpha\eta \max_x \frac{q(x)}{\mathcal{D}_X(x)} \gtrsim \frac{\alpha}{m^*\left(H, \mathcal{D}_X, c_4\eta, c_5\varepsilon - 2\eta, \frac{99}{100}\right)} \quad if \quad h^* \notin S_i,$$

*as well as $|\Delta_i| \leq \alpha$ always and $\mathrm{Var}[\Delta_i | \mathcal{F}_i] \leq \alpha \mathbb{E}\left[|\Delta_i| | \mathcal{F}_i\right] \lesssim \alpha \mathbb{E}[\Delta_i | \mathcal{F}_i]$.*

*Proof.* For the sake of bookkeeping, we let $m^* = m^*\left(H, \mathcal{D}_X, c_4\eta, c_5\varepsilon - 2\eta, \frac{99}{100}\right)$ in this proof and the following text. We first bound the expectation. By Lemma A.1 applied to $S \in \{H, H \setminus S_i\}$ with $3\eta$, we have

$$\mathbb{E}\left[\Delta_i | \mathcal{F}_i\right] - 2\alpha\eta \max_x \frac{q(x)}{\mathcal{D}_X(x)} \geq 0.9\alpha \left(\mathbb{E}_{x \sim q}\left[r_{i,H,h^*}(x) + r_{i,H \setminus S_i, h^*}(x)\right] - 13.8\eta \max_x \frac{q(x)}{\mathcal{D}_X(x)}\right)$$
$$- 2\alpha\eta \max_x \frac{q(x)}{\mathcal{D}_X(x)},$$

where $q$ is the query distribution of the algorithm at iteration $i$. Now, by the definition of

$$\overline{\lambda}_{i,S} = \frac{1}{2}\lambda_i + \frac{1}{2}\lambda_{i,H \setminus S},$$

we have for any $x$ that

$$\overline{r}_{i,S_i,h^*}(x) = \frac{1}{2}(r_{i,h^*}(x) + r_{i,H \setminus S_i, h^*}(x))$$

and thus

$$\mathbb{E}\left[\Delta_i | \mathcal{F}_i\right] - 2\alpha\eta \max_x \frac{q(x)}{\mathcal{D}_X(x)}$$
$$\geq 1.8\alpha \left(\mathbb{E}_{x \sim q}\left[\overline{r}_{i,S_i,h^*}(x)\right] - 6.9\eta \max_x \frac{q(x)}{\mathcal{D}_X(x)}\right) - 2\alpha\eta \max_x \frac{q(x)}{\mathcal{D}_X(x)} \tag{7}$$
$$\geq 1.8\alpha \left(\mathbb{E}_{x \sim q}\left[\overline{r}_{i,S_i}(x)\right] - 8.1\eta \max_x \frac{q(x)}{\mathcal{D}_X(x)}\right).$$

Algorithm 1 chooses the sampling distribution $q$ to maximize $\mathbb{E}_{x \sim q}\left[\overline{r}_{i,S_i}(x)\right] - \frac{c_4}{20}\eta \max_x \frac{q(x)}{\mathcal{D}_X(x)} \leq \mathbb{E}_{x \sim q}\left[\overline{r}_{i,S_i}(x)\right] - 15\eta \max_x \frac{q(x)}{\mathcal{D}_X(x)}$ because $c_4 \geq 300$. By Lemma A.2, $\overline{\lambda}_{i,S_i}$ over $H'$ has no radius-$(c_4\eta + c_5\varepsilon)$ ball with probability larger than $80\%$, so by Lemma 2.1 $q$ satisfies

$$\mathbb{E}_{x \sim q}\left[\overline{r}_{i,S_i}(x)\right] - 15\eta \max_x \frac{q(x)}{\mathcal{D}_X(x)} \geq \mathbb{E}_{x \sim q}\left[\overline{r}_{i,S_i}(x)\right] - \frac{c_4}{20}\eta \max_x \frac{q(x)}{\mathcal{D}_X(x)} \gtrsim \frac{1}{m^*\left(H', \mathcal{D}_X, c_4\eta, c_5\varepsilon, \frac{99}{100}\right)}.$$

Because $H' \subseteq H$ is a maximal $2\eta$-packing, every hypothesis in $H$ is within $2\eta$ of some hypothesis in $H'$. The problem $\left(H, \mathcal{D}_X, c_4\eta, c_5\varepsilon - 2\eta, \frac{99}{100}\right)$ is harder than the problem $\left(H', \mathcal{D}_X, c_4\eta, c_5\varepsilon, \frac{99}{100}\right)$ because we can reduce the latter to the former by simply adding more hypotheses and solve

it then map the solution back by returning the closest hypothesis in $H'$. Hence, $m^* \geq m^*\left(H', \mathcal{D}_X, c_4\eta, c_5\varepsilon, \frac{99}{100}\right)$. Therefore,

$$\mathbb{E}\left[\Delta_i | \mathcal{F}_i\right] - 2\alpha\eta \max_x \frac{q(x)}{\mathcal{D}_X(x)} \geq 1.8\alpha \left(\underset{x \sim q}{\mathbb{E}}\left[\overline{r}_{i,S_i}(x)\right] - 8.1\eta \max_x \frac{q(x)}{\mathcal{D}_X(x)}\right) \gtrsim \frac{\alpha}{m^*}.$$

We now bound the variance. The value of $\Delta_i$ may be positive or negative, but it is bounded by $|\Delta_i| \leq \alpha$. Thus

$$\text{Var}[\Delta_i | \mathcal{F}_i] \leq \mathbb{E}\left[\Delta_i^2 | \mathcal{F}_i\right] \leq \alpha \, \mathbb{E}[|\Delta_i| \,|\mathcal{F}_i].$$

By Lemma A.1 and (7) we have

$$\mathbb{E}[|\Delta_i| \,|\mathcal{F}_i] = \mathbb{E}[2\max\{\Delta_i, 0\} - \Delta_i | \mathcal{F}_i]$$

$$\leq 4\alpha \underset{x \sim q}{\mathbb{E}}\left[\overline{r}_{i,S_i}(x)\right] - 1.8\alpha \left(\underset{x \sim q}{\mathbb{E}}\left[\overline{r}_{i,S_i}(x)\right] - 8.1\eta \max_x \frac{q(x)}{\mathcal{D}_X(x)}\right)$$

$$\leq 2.2\alpha \left(\underset{x \sim q}{\mathbb{E}}\left[\overline{r}_{i,S,h^*}(x)\right] + 6.7\eta \max_x \frac{q(x)}{\mathcal{D}_X(x)}\right)$$

$$\leq \frac{2.2\alpha}{1.8\alpha} \mathbb{E}\left[\Delta_i | \mathcal{F}_i\right] + 2.2\alpha \cdot 6.9\eta \max_x \frac{q(x)}{\mathcal{D}_X(x)} + 2.2\alpha \cdot 6.7\eta \max_x \frac{q(x)}{\mathcal{D}_X(x)}$$

$$\leq 1.3 \, \mathbb{E}[\Delta_i | \mathcal{F}_i] + 30\alpha\eta \max_x \frac{q(x)}{\mathcal{D}_X(x)}.$$

Since $\mathbb{E}_{x \sim q}[\Delta_i | \mathcal{F}_i] - 2\alpha\eta \max_x \frac{q(x)}{\mathcal{D}_X(x)} \gtrsim \frac{1}{m^*} \geq 0$, we have

$$\eta \max_x \frac{q(x)}{\mathcal{D}_X(x)} \leq \frac{1}{2\alpha} \underset{x \sim q}{\mathbb{E}}\left[\Delta_i | \mathcal{F}_i\right],$$

and thus

$$\text{Var}[\Delta_i | \mathcal{F}_i] \leq \alpha \, \mathbb{E}[|\Delta_i| \,|\mathcal{F}_i] \lesssim \alpha \, \mathbb{E}\left[\Delta_i | \mathcal{F}_i\right].$$

$\square$

## A.3 Concentration of potential

We have showed that the potential will grow at $\Omega\left(\frac{1}{m^*}\right)$ per iteration, but only in expectation, while our goal is to obtain a high probability bound. Let $\mu_k := \sum_{i<k} \mathbb{E}[\Delta_i | \mathcal{F}_{i-1}] \gtrsim k/m^*$, then

$$\mathbb{E}\left[(\psi_k - \mu_k) - (\psi_{k-1} - \mu_{k-1}) | \mathcal{F}_{k-1}\right] = \mathbb{E}\left[\psi_k - \psi_{k-1} | \mathcal{F}_{k-1}\right] - (\mu_k - \mu_{k-1})$$
$$= \mathbb{E}\left[\Delta_i | \mathcal{F}_i\right] - \mathbb{E}\left[\Delta_i | \mathcal{F}_i\right] \geq 0.$$

Apparently $|\psi_k - \mu_k|$ is upper bounded, so $\psi_k - \mu_k$ is a supermartingale. To show a high probability bound, we will use Freedman's inequality. A version is stated in Tropp [2011]. We slighted modify it so it can be applied to supermartingale as the following. (XXX Not sure if the following is correct. I can't find a version of supermartingale.)

**Theorem A.5** (Freedman's Inequality). *Consider a real-valued supermartingale $\{Y_k : k = 0, 1, 2, \cdots\}$ that is adapted to the filtration $\mathcal{F}_0 \subseteq \mathcal{F}_1 \subseteq \mathcal{F}_2 \subseteq \cdots \subseteq \mathcal{F}$ with difference sequence $\{X_k : k = 1, 2, 3, \cdots\}$. Assume that the difference sequence is uniformly bounded:*

$$X_k \leq R \text{ almost surely for } k = 1, 2, 3, \cdots$$

*Define the predictable quadratic variation process of the supermartingale:*

$$W_k := \sum_{j=1}^{k} \mathbb{E}\left[X_j^2 | \mathcal{F}_{j-1}\right] \text{ for } k = 1, 2, 3, \cdots$$

*Then, for all $t \geq 0$ and $\sigma^2 > 0$,*

$$\Pr\left(\exists k \geq 0 : Y_k \leq -t \text{ and } W_k \leq \sigma^2\right) \leq \exp\left(-\frac{t^2/2}{\sigma^2 + Rt/3}\right).$$

Then we can prove a high probability bound as the following.

**Lemma A.6.** *With probability $1 - \delta$, $\phi_i = 0$ for some $i = O\left(m^* \log \frac{|H|}{\delta}\right)$ so $h^* \in S_i$.*

*Proof.* Remember we have that

$$\phi_k = \phi_0 + \psi_k + \sum_{i<k} \log \frac{\lambda_{i,H\setminus S_{i+1}}(h^*)}{\lambda_{i,H\setminus S_i}(h^*)}.$$

Since $S_{i+1} \supseteq S_i$ for all $i$, $\lambda_{i,H\setminus S_{i+1}}(h^*) \geq \lambda_{i,H\setminus S_i}(h^*)$ if $h^* \notin S_{i+1}$, we have

$$\phi_k \geq \phi_0 + \psi_k \quad \text{if } h^* \notin S_k.$$

Let $K = O\left(m^* \log \frac{|H|}{\delta}\right)$. Let's assume by contradiction that $\phi_K < 0$ for for, then $h^* \notin S_i$ for $i \leq K$. We know by Lemma A.4 that

$$\mu_k := \sum_{i<k} \mathbb{E}[\Delta_i|\mathcal{F}_{i-1}] \gtrsim \frac{k}{m^*}$$

and that $\sum_{i<k} \text{Var}\,[\Delta_i] \leq \frac{1}{4}\mu_k$ by picking $\alpha$ small enough. Moreover, $|\Delta_i| \leq \alpha$ always. To use Freedman's inequality, let's set the RHS

$$\exp\left(-\frac{t^2/2}{\sigma^2 + Rt/3}\right) \leq \delta.$$

Solving the above quadratic equation, one solution is that $t \geq \frac{R}{3}\log\frac{1}{\delta} + \sqrt{\frac{R^2}{9}\log^2\frac{1}{\delta} + 2\sigma^2\log\frac{1}{\delta}}$. Let's substitute in $R = \alpha$ and $\sigma^2 = \sum_{i<k} \text{Var}_{i-1}(\Delta_i)$, with $1 - \delta$ probability we have for any $k > O(m^* \log \frac{1}{\delta})$ that

$$\psi_k \geq \mu_k - \sqrt{\frac{\alpha^2}{9}\log^2\frac{1}{\delta} + 2\sum_{i<k}\text{Var}_{i-1}(\Delta_i)\log\frac{1}{\delta}} - \frac{\alpha}{3}\log\frac{1}{\delta}$$

$$\geq \mu_k - \sqrt{\frac{\alpha^2}{9}\log^2\frac{1}{\delta} + \frac{1}{2}\mu_k\log\frac{1}{\delta}} - \frac{\alpha}{3}\log\frac{1}{\delta}$$

$$\geq \mu_k - \max\left\{\frac{\sqrt{2}\alpha}{3}\log\frac{1}{\delta}, \sqrt{\mu_k\log\frac{1}{\delta}}\right\} - \frac{\alpha}{3}\log\frac{1}{\delta}$$

$$\geq \frac{1}{2}\mu_k$$

$$\gtrsim \frac{k}{m^*}.$$

The second last inequality is because the first term outscales all of the rest. Since $K = O\left(m^* \log \frac{|H|}{\delta}\right)$, we have

$$\psi_K \geq 2\log|H|$$

with $1 - \delta$ probability. Then $\phi_K \geq \phi_0 + \psi_k$ because $\phi_0 \geq \log\frac{1}{2|H|} \geq -2\log|H|$ and this contradicts $h^* \notin S_K$. Therefore, with probability at least $1 - \delta$, $h^* \in S_K$ and by definition, $\phi_i = 0$ for some $i \leq K$ as desired.

$\square$

### A.4 Bounding the Size of $|C|$

So far we've shown that after $O\left(m^* \log \frac{|H|}{\delta}\right)$ iterations, $h^*$ will be included in the set $S_i$. The last thing we need to prove Theorem 1.1 is that with high probability, $C$ is small, which is equivalent to show that not many balls will be added to $S_i$ after $O\left(m^* \log \frac{|H|}{\delta}\right)$ iterations. To show this, we first need to relate the number of balls added to $S_i$ to $\psi_i$. Let $\mathcal{E}_i$ denote the number of errors $h^*$ made up to iteration $i$ (and set $\mathcal{E}_i = \mathcal{E}_{i-1}$ if $h^* \in S_i$) and $\mathcal{N}_i$ denote the number of balls added to $S_i$ up to iteration $i$ (again set $\mathcal{N}_i = \mathcal{N}_{i-1}$ if $h^* \in S_i$).

**Lemma A.7.** *The following inequality holds for every $i$:*

$$\mathcal{N}_i \leq 5(\psi_i + 2\alpha\mathcal{E}_i) + 1.$$

*Proof.* We divide the $i$ iterations into phases. A new phase begins and an old phase ends if at this iteration a new ball is added to the set $S_i$. We use $p_1, \ldots, p_k$ for $k \leq i$ to denote phases and $i_1, \ldots, i_k$ to denote the starting iteration of the phases. We analyse how the potential changes from the phase $p_j$ to the phase $p_{j+1}$. Let's say the ball $B_2 = (\mu_2, 3c_4\eta + 3c_5\varepsilon)$ is added at the beginning of $p_{j+1}$ and $B_1 = (\mu_1, 3c_4\eta + 3c_5\varepsilon)$ is the ball added at the beginning of $p_j$. Then the ball $B_2' = (\mu_2, c_4\eta + c_5\varepsilon)$ and the ball $B_1' = (\mu_1, c_4\eta + c_5\varepsilon)$ are disjoint. Otherwise, $B_2' \subseteq B_1$ so $B_2$ would not have been added by the algorithm. At the beginning of $p_j$, $B_1'$ has probability no less than $0.6$ by Claim A.3. Therefore, $B_2'$ has probability no more than $0.4$. Similarly, at the beginning of $p_{j+1}$, $B_2'$ has probability at least $0.6$ by Claim A.3. Since during one iteration the weight of a hypothesis cannot change too much, at iteration $i_{j+1} - 1$, $B_2'$ has weight at least $0.5$ by picking $\alpha$ small enough. Therefore, we have $\log \lambda_{i_{j+1}-1}(B_2') - \log \lambda_{i_j}(B_2') \geq \log \frac{0.5}{0.4} \geq \frac{1}{5}$. Moreover, note that $S_i$ does not change from iteration $i_j$ to iteration $i_{j+1} - 1$ by the definition of phases. Now we compute

$$\sum_{l=i_j}^{i_{j+1}-1} \Delta_l = \log \frac{\lambda_{i_{j+1}-1}(h^*)}{\lambda_{i_j}(h^*)} + \log \frac{\lambda_{i_{j+1}-1, H\backslash S_{i_j}}(h^*)}{\lambda_{i_j, H\backslash S_{i_j}}(h^*)},$$

$$= \log \frac{w_{i_{j+1}-1}(h^*)}{w_{i_1}(h^*)} \frac{\sum_{h\in H} w_{i_1}(h)}{\sum_{h\in H} w_{i_{j+1}-1}(h)} + \log \frac{w_{i_{j+1}-1}(h^*)}{w_{i_j}(h^*)} \frac{w_{i_j}(H\backslash S_{i_j})}{w_{i_{j+1}-1}(H\backslash S_{i_j})}.$$

The change of the weight of $h^*$ is

$$\frac{w_{i_{j+1}}(h^*)}{w_{i_j}(h^*)} = e^{-\alpha\mathcal{E}_{p_j}},$$

where $\mathcal{E}_{p_j}$ is the number of errors $h^*$ made in $p_j$. Consequently,

$$\sum_{l=i_j}^{i_{j+1}-1} \Delta_l = -2\alpha\mathcal{E}_{p_j} + \log \frac{\sum_{h\in H} w_{i_j}(h)}{\sum_{h\in H} w_{i_{j+1}-1}(h)} + \log \frac{w_{i_j}(H\backslash S_{i_j})}{w_{i_{j+1}-1}(H\backslash S_{i_j})}$$

$$\geq -2\alpha\mathcal{E}_{p_j} + \frac{1}{5}.$$

The last step above comes from

$$\log \frac{\sum_{h\in H} w_{i_j}(h)}{\sum_{h\in H} w_{i_{j+1}-1}(h)} \geq \log \frac{\sum_{h\in B_2'} w_{i_{j+1}-1}(h)}{\sum_{h\in B_2'} w_{i_j}(h)} \frac{\sum_{h\in H} w_{i_j}(h)}{\sum_{h\in H} w_{i_{j+1}-1}(h)} = \log \frac{\lambda_{i_{j+1}-1}(B_2')}{\lambda_{i_j}(B_2')} \geq \frac{1}{5},$$

and

$$\log \frac{w_{i_j}(H\backslash S_{i_j})}{w_{i_{j+1}-1}(H\backslash S_{i_j})} \geq 0$$

because the weight $w(h)$ only decreases. Summing over all phases $j$ and we get

$$\psi_i \geq -2\alpha\mathcal{E}_i + \frac{1}{5}(\mathcal{N}_i - 1).$$

Since $i$ may not exactly be the end of a phase, the last phase may end early so we have $\mathcal{N}_i - 1$ instead of $\mathcal{N}_i$. Rearrange and the proof finishes. $\square$

We have already bounded $\psi_i$, so we just need to bound $\mathcal{E}_i$ in order to bound $\mathcal{N}_i$ by the following lemma.

**Lemma A.8.** *For every $k$, with probability at least $1 - \delta$,*

$$\mathcal{E}_k \leq \frac{1}{\alpha}\left(\psi_k + \sqrt{2}\log\frac{1}{\delta}\right).$$

*Proof.* Let $q$ be the query distribution at iteration $i-1$ and $p(x)$ be the probability that $x$ is corrupted by the adversary. Then the conditional expectation of $\mathcal{E}_i - \mathcal{E}_{i-1}$ is

$$\mathbb{E}\left[\mathcal{E}_i - \mathcal{E}_{i-1}|\mathcal{F}_i\right] = \Pr_{x \sim q}\left[h^*(x) \text{ is wrong}\right] = \mathbb{E}_{x \sim q}\left[p(x)\right] = \mathbb{E}_{x \sim \mathcal{D}}\left[p(x)\frac{q(x)}{\mathcal{D}(x)}\right] \leq \eta \max_x \frac{q(x)}{\mathcal{D}_X(x)}.$$

Then if $h^* \notin S$, from Lemma A.4

$$\mathbb{E}[\Delta_i - 2\alpha(\mathcal{E}_i - \mathcal{E}_{i-1})|\mathcal{F}_i] \geq \mathbb{E}\left[\Delta_i|\mathcal{F}_i\right] - 2\alpha\eta \max_x \frac{q(x)}{\mathcal{D}_X(x)} \gtrsim \frac{1}{m^*}.$$

Therefore, $\mathbb{E}[\alpha\left(\mathcal{E}_i - \mathcal{E}_{i-1}\right)|\mathcal{F}_i] \leq \frac{1}{2}\mathbb{E}\left[\Delta_i|\mathcal{F}_i\right]$ and $\mathbb{E}[\Delta_i - \alpha(\mathcal{E}_i - \mathcal{E}_{i-1})|\mathcal{F}_i] \geq \frac{1}{2}\mathbb{E}\left[\Delta_i|\mathcal{F}_i\right]$. This means that $\psi_k - \alpha\mathcal{E}_k - \frac{1}{2}\mu_k$ is a supermartingale. We then bound $\text{Var}\left[\Delta_i - \alpha(\mathcal{E}_i - \mathcal{E}_{i-1})|\mathcal{F}_i\right]$. Note that $|\Delta_i - \alpha(\mathcal{E}_i - \mathcal{E}_{i-1})| \leq 2\alpha$, so

$$\text{Var}[\Delta_i - \alpha(\mathcal{E}_i - \mathcal{E}_{i-1})|\mathcal{F}_i] \leq \mathbb{E}\left[(\Delta_i - \alpha(\mathcal{E}_i - \mathcal{E}_{i-1}))^2 \Big| \mathcal{F}_i\right] \leq 2\alpha\,\mathbb{E}\left[|\Delta_i - \alpha(\mathcal{E}_i - \mathcal{E}_{i-1})| \,|\mathcal{F}_i\right].$$

Furthermore,

$$\mathbb{E}\left[|\Delta_i - \alpha(\mathcal{E}_i - \mathcal{E}_{i-1})| \,|\mathcal{F}_i\right] \leq \mathbb{E}\left[|\Delta_i||\mathcal{F}_i\right] + \alpha\eta \max_x \frac{q(x)}{\mathcal{D}_X(x)}.$$

As a result,

$$\text{Var}[\Delta_i - \alpha(\mathcal{E}_i - \mathcal{E}_{i-1})|\mathcal{F}_i] \leq 2\alpha\left(\mathbb{E}\left[|\Delta_i||\mathcal{F}_i\right] + \alpha\eta \max_x \frac{q(x)}{\mathcal{D}_X(x)}\right)$$

$$\leq 2\alpha\left(\mathbb{E}\left[|\Delta_i||\mathcal{F}_i\right] + \frac{1}{2}\mathbb{E}\left[\Delta_i|\mathcal{F}_i\right]\right)$$

$$\leq 3\alpha\,\mathbb{E}\left[|\Delta_i||\mathcal{F}_i\right]$$

$$\lesssim \alpha\,\mathbb{E}\left[\Delta_i|\mathcal{F}_i\right].$$

By picking $\alpha$ small enough, $\sum_{i<k} \text{Var}\left[\Delta_i - \alpha(\mathcal{E}_i - \mathcal{E}_{i-1})|\mathcal{F}_i\right] \leq \frac{1}{8}\mu_k$. Moreover, $|\Delta_i - \alpha(\mathcal{E}_i - \mathcal{E}_{i-1})| \leq 2\alpha$ always. Therefore by Freedman's inequality, with $1 - \delta$ probability we have for any $k$ that

$$\psi_k - \alpha\mathcal{E}_k \geq \mu_k - \sqrt{\frac{4\alpha^2}{9}\log^2\frac{1}{\delta} + 2\sum_{i<k}\text{Var}_{i-1}\left[\Delta_i - \alpha(\mathcal{E}_i - \mathcal{E}_{i-1})\right]\log\frac{1}{\delta}} - \frac{2\alpha}{3}\log\frac{1}{\delta}$$

$$\geq \mu_k - \sqrt{\frac{4\alpha^2}{9}\log^2\frac{1}{\delta} + \frac{1}{4}\mu_k\log\frac{1}{\delta}} - \frac{2\alpha}{3}\log\frac{1}{\delta}$$

$$\geq \mu_k - \max\left\{\frac{2\sqrt{2}\alpha}{3}\log\frac{1}{\delta}, \frac{\sqrt{2}}{2}\sqrt{\mu_k\log\frac{1}{\delta}}\right\} - \frac{2\alpha}{3}\log\frac{1}{\delta}$$

$$\geq \mu_k - \max\left\{\frac{\sqrt{2}}{2}\log\frac{1}{\delta}, \frac{\sqrt{2}}{2}\mu_k\right\} - \frac{2\alpha}{3}\log\frac{1}{\delta}$$

$$\geq \left(1 - \frac{\sqrt{2}}{2}\right)\mu_k - \sqrt{2}\log\frac{1}{\delta}$$

$$\geq -\sqrt{2}\log\frac{1}{\delta}$$

Rearrange and we proved the lemma. $\qquad\square$

Combining Lemma A.7 and Lemma A.8, we can show $C$ is small with high probability as the lemma follows.

**Lemma A.9.** *For* $k = O\left(m^* \log \frac{|H|}{\delta}\right)$, *with probability at least* $1 - 2\delta$, $h^* \in S_k$ *and* $|C| \leq O\left(\log \frac{|H|}{\delta}\right)$ *at iteration* $k$.

*Proof.* By union bound, with probability at least $1 - 2\delta$, Lemma A.6 and A.8 will hold at the same time. This means $h^*$ is added to $S_k$. By definition, $0 \geq \phi_k \geq \phi_0 + \psi_k$, so $\psi_k \leq 2\log|H|$. Therefore, by Lemma A.7 and A.8, the number of balls added $|C|$ is $O\left(\log|H| + \log\frac{1}{\delta}\right) = O\left(\log\frac{|H|}{\delta}\right)$. $\quad\square$

## A.5 Putting Everything Together

We proved that after $O\left(m^* \log \frac{|H|}{\delta}\right)$ iterations, $h^* \in S_i$ and $C$ is small with high probability. Hence, running the stage two algorithm to return a desired hypothesis will not take much more queries. We are ready to put everything together and finally prove Theorem 1.1.

**Theorem 1.1** (Competitive Bound). *There exist some constants $c_1, c_2$ and $c_3$ such that for any instance $(H, \mathcal{D}_X, \eta, \varepsilon, \delta)$ with $\varepsilon \geq c_1 \eta$, Algorithm 1 solves the instance with sample complexity*

$$m(H, \mathcal{D}_X, \eta, \varepsilon, \delta) \lesssim \left( m^* \left( H, \mathcal{D}_X, c_2\eta, c_3\varepsilon, \frac{99}{100} \right) + \log \frac{1}{\delta} \right) \cdot \log \frac{N(H, \mathcal{D}_X, \eta)}{\delta}$$

*and polynomial time.*

*Proof.* Let's pick $c_1, c_4, c_5$ as in Theorem 2.3 and pick the confidence parameter to be $\frac{\delta}{3}$. Then by Lemma A.9, with probability $1 - \frac{2\delta}{3}$, the first $O\left(\log \frac{|H|}{\delta}\right)$ ball added to $S_i$ will contain $h^*$. Since each ball added to $C$ has radius $3c_4\eta + 3c_5\varepsilon$, the best hypothesis in $C$ has error $(3 + 3c_4)\eta + 3c_5\varepsilon$. By Theorem 2.2, with probability $1 - \frac{\delta}{3}$, the algorithm will return a hypothesis with error $(9 + 9c_4)\eta + 9c_5\varepsilon \leq \eta + \varepsilon$. Therefore, by union bound, the algorithm will return a desired hypothesis with probability $1 - \delta$. This proves the correctness of the algorithm.

The stage one algorithm makes

$$O\left( m^* \left( H, \mathcal{D}_X, c_4\eta, c_5\varepsilon - 2\eta, \frac{99}{100} \right) \log \frac{|H|}{\delta} \right) \leq O\left( m^* \left( H, \mathcal{D}_X, c_4\eta, \frac{c_5}{2}\varepsilon, \frac{99}{100} \right) \log \frac{|H|}{\delta} \right)$$

queries. The stage two algorithm makes $O\left(|C| \log \frac{|C|}{\delta}\right)$ queries by Theorem 2.2. Note that $C$ is a $c_4\eta + c_5\varepsilon$-packing because the center of added balls are at least $c_4\eta + c_5\varepsilon$ away, so $m^*\left(H, \mathcal{D}_X, \frac{c_4}{2}\eta, \frac{c_5}{2}\varepsilon, \frac{99}{100}\right) \geq \log |C|$. Since $|C| \leq \log \frac{|H|}{\delta}$ by Lemma A.9, stage two algorithm takes $O\left( \left( m^* \left( H, \mathcal{D}_X, \frac{c_4}{2}\eta, \frac{c_5}{2}\varepsilon, \frac{99}{100} \right) + \log \frac{1}{\delta} \right) \log \frac{|H|}{\delta} \right)$ queries. Picking $c_2 = c_4, c_3 = \frac{c_5}{2}$, we get the desired sample complexity bound.

To compute the packing at the beginning of the algorithm, we need to compute the distance of every pair of hypotheses, which takes $O(|H|^2 |\mathcal{X}|)$ time. Computing $r$ in each round takes $O(|H||\mathcal{X}|)$ time and solving the optimization problem takes $O(|\mathcal{X}|)$ time. Therefore, the remaining steps in stage one takes $O\left(m^*|H||\mathcal{X}| \log \frac{|H|}{\delta}\right)$ time. Stage two takes $O\left(\log \frac{|H|}{\delta} \log \frac{\log \frac{|H|}{\delta}}{\delta}\right)$ time. Therefore, the overall running time is polynomial of the size of the problem. $\square$

Similarly, we can prove Theorem 2.3, which is a stronger and more specific version of Theorem 1.1.

**Theorem 2.3.** *Suppose that $\mathcal{D}_x$ and $H$ are such that, for any distribution $\lambda$ over $H$ such that no radius-$(c_4\eta + c_5\varepsilon)$ ball has probability more than $80\%$, there exists a distribution $q$ over $X$ such that*

$$\mathbb{E}_{x \sim q}[r(x)] - \frac{c_4}{20}\eta \max_x \frac{q(x)}{\mathcal{D}_x(x)} \geq \beta$$

*for some $\beta > 0$. Then for $\varepsilon \geq c_1\eta$, $c_4 \geq 300$, $c_5 = \frac{1}{10}$ and $c_1 \geq 90c_4$, let $N = N(H, \mathcal{D}_x, \eta)$ be the size of an $\eta$-cover of $H$. Algorithm 1 solves $(\eta, \varepsilon, \delta)$ active agnostic learning with $O\left(\frac{1}{\beta} \log \frac{N}{\delta} + \log \frac{N}{\delta} \log \frac{\log N}{\delta}\right)$ samples.*

*Proof.* By Lemma A.9 (with $m^*$ replaced by $\frac{1}{\beta}$ and setting confidence parameter to $\frac{\delta}{3}$) after $O\left(\frac{1}{\beta} \log \frac{N}{\delta}\right)$ queries, with probability at least $1 - \frac{2\delta}{3}$, a hypothesis in $C$ will be within $c_4\eta + c_5\varepsilon$ to $h^*$ and $|C| = O\left(\log \frac{N}{\delta}\right)$. From Theorem 2.2, with probability at least $1 - \frac{\delta}{3}$, stage two algorithm then outputs a hypothesis $\hat{h}$ that is $9c_4\eta + 9c_5\varepsilon$ from $h'$ so $\mathrm{err}\left(\hat{h}\right) \leq 9c_4\eta + 9c_5\varepsilon \leq \eta + \varepsilon$ by the choice of the constants. The stage two algorithm makes $O\left(\log \frac{N}{\delta} \log \frac{\log \frac{N}{\delta}}{\delta}\right)$ queries. Overall, the algorithm makes $O\left(\frac{1}{\beta} \log \frac{N}{\delta} + \log \frac{N}{\delta} \log \frac{\log \frac{N}{\delta}}{\delta}\right)$ queries and succeeds with probability at least $1 - \delta$. $\square$

# B  Query Complexity Lower Bound

In this section we derive a lower bound for the agnostic binary classification problem, which we denote by AGNOSTICLEARNING. The lower bound is obtained from a reduction from minimum set cover, which we denote by SETCOVER. The problem SETCOVER consists a pair $(U, \mathcal{S})$, where $U$ is a ground set and $\mathcal{S}$ is a collection of subsets of $U$. The goal is to find a set cover $C \subseteq \mathcal{S}$ such that $\bigcup_{S \in C} S = U$ of minimal size $|C|$. We use $K$ to denote the cardinality of the minimum set cover.

**Lemma B.1** (Dinur and Steurer [2014], Corollary 1.5). *There exists hard instances* SETCOVER-HARD *with the property* $K \geq \log |U|$ *such that for every* $\gamma > 0$, *it is NP-hard to approximate* SETCOVERHARD *to within* $(1 - \gamma) \ln |U|$.

*Proof.* This lemma directly follows from Dinur and Steurer [2014, Corollary 1.5]. In their proof, they constructed a hard instance of SETCOVER from LABELCOVER. The size of the minimum cover $K \geq |V_1| = D n_1$ and $\log |U| = (D + 1) \ln n_1 \leq K$. So the instance in their proof satisfies the desired property. $\qquad \square$

Then we prove the following lemma by giving a ratio-preserving reduction from SETCOVER to AGNOSTICLEARNING.

**Lemma B.2.** *If there exists a deterministic $\alpha$-approximation algorithm for* AGNOSTICLEARNING $\left(H, \mathcal{D}_x, \frac{1}{3|\mathcal{X}|}, \frac{1}{3|\mathcal{X}|}, \frac{1}{4|H|}\right)$, *there exists a deterministic $2\alpha$-approximation algorithm for* SETCOVERHARD.

*Proof.* Given an instance of SETCOVERHARD, for each $s \in \mathcal{S}$, number the elements $u \in s$ in an arbitrary order; let $f(s, u)$ denote the index of $u$ in $s$'s list (and padding 0 to the left with the extra bit). We construct an instance of AGNOSTICLEARNING as the following:

1. Let the domain $\mathcal{X}$ have three pieces: $U$, $V := \{(s, j) \mid s \in \mathcal{S}, j \in [1 + \log |s|]\}$, and $D = \{1, \ldots, \log |U|\}$, an extra set of $\log |U|$ more coordinates.

2. On this domain, we define the following set of hypotheses:

    (a) For $u \in U$, define $h_u$ which only evaluates 1 on $u \in U$ and on $(s, j) \in V$ if $u \in s$ and the $j$'th bit of $(2f(s, u) + 1)$ is 1.

    (b) For $d \in D$, define $h_d$ which only evaluates 1 on $d$.

    (c) Define $h_0$ which evaluates everything to 0.

3. Let $\mathcal{D}_X$ be uniform distribution over $\mathcal{X}$ and set $\eta = \frac{1}{3|\mathcal{X}|}$ and $\varepsilon = \frac{1}{3|\mathcal{X}|}$. Set $\delta = \frac{1}{4|H|}$.

Any two hypotheses satisfy $\|h_1 - h_2\| \geq \frac{1}{|\mathcal{X}|} > \varepsilon = \eta$, so $\mathrm{err}(h^*) = 0$. First we show that $m^* \left(H, \mathcal{D}_x, \frac{1}{3|\mathcal{X}|}, \frac{1}{3|\mathcal{X}|}, \frac{1}{4|H|}\right) \leq K + \log |U|$. Indeed there exists a deterministic algorithm using $K + \log |U|$ queries to identify any hypothesis with probability 1. Given a smallest set cover $C$, the algorithm first queries all $(s, 0) \in V$ for $s \in C$. If $h^* = h_u$ for some $u$, then for the $s \in \mathcal{S}$ that covers $u$, $(s, 0)$ will evaluate to true. The identity of $u$ can then be read out by querying $(s, j)$ for all $j$. The other possibilities–$h_d$ for some $d$ or 0—can be identified by evaluating on all of $D$ with $\log U$ queries. The total number of queries is then at most $K + \log |U|$ in all cases, so $m^* \leq K + \log |U| \leq 2K$.

We now show how to reconstruct a good approximation to set cover from a good approximate query algorithm. We feed the query algorithm $y = 0$ on every query it makes, and let $C$ be the set of all $s$ for which it queries $(s, j)$ for some $j$. Also, every time the algorithm queries some $u \in U$, we add an arbitrary set containing $u$ to $C$. Then the size of $C$ is at most the number of queries. We claim that $C$ is a set cover: if $C$ does not cover some element $u$, then $h_u$ is zero on all queries made by the algorithm, so $h_u$ is indistinguishable from $h_0$ and the algorithm would fail on either input $h_0$ or $h_u$. Thus if $A$ is a deterministic $\alpha$-approximation algorithm for AGNOSTICLEARNING, we will recover a set cover of size at most $\alpha m^* \leq \alpha (K + \log |U|) \leq 2\alpha K$, so this gives a deterministic $2\alpha$-approximation algorithm for SETCOVERHARD. $\qquad \square$

Similar results also holds for randomized algorithms, we just need to be slightly careful about probabilities.

**Lemma B.3.** *If there exists a randomized algorithm for* AGNOSTICLEARNING $\left(H, \mathcal{D}_x, \frac{1}{3|\mathcal{X}|}, \frac{1}{3|\mathcal{X}|}, \frac{1}{4|H|}\right)$, *there exists a randomized $2\alpha$-approximation algorithm for* SETCOVERHARD *with success probability at least $\frac{2}{3}$.*

*Proof.* We use the same reduction as in Lemma B.2. Let $A$ be an algorithm solves AGNOSTICLEARNING $\left(H, \mathcal{D}_x, \frac{1}{3|\mathcal{X}|}, \frac{1}{3|\mathcal{X}|}, \frac{1}{4|H|}\right)$. To obtain a set cover using $A$, we keeping giving $A$ label 0 and construct the set $C$ as before. Let $q_C$ be a distribution over the reconstructed set $C$. Assume that by contradiction with probability at least $\frac{1}{3}$, $C$ is not a set cover. Then, with probability at least $1/3$, there is some element $v$ such that both $h_v$ and $h_0$ are consistent on all queries the algorithm made; call such a query set "ambiguous".

Then what is the probability that the agnostic learning algorithm fails on the input distribution that chooses $h^*$ uniformly from $H$? Any given ambiguous query set is equally likely to come from any of the consistent hypotheses, so the algorithm's success probability on ambiguous query sets is at most $1/2$. The chance the query set is ambiguous is at least $\frac{2}{3|H|}$: a $\frac{1}{3H}$ chance that the true $h^*$ is $h_0$ and the query set is ambiguous, and at least as much from the other hypotheses making it ambiguous. Thus the algorithm's fails to learn the true hypothesis with at least $\frac{1}{3|H|}$ probability, contradicting the assumed $\frac{1}{4|H|}$ failure probability.

Therefore, a set cover of size at most $2\alpha K$ can be recovered with probability at least $\frac{1}{3}$ using the agnostic learning approximation algorithm. $\square$

The following theorem will then follow.

**Theorem 1.2** (Lower Bound). *It is NP-hard to find a query strategy for every agnostic active learning instance within an $c \log |H|$ for some constant $c > 0$ factor of the optimal sample complexity.*

*Proof.* Let's consider the instance of set cover constructed in Lemma B.2. Let $c = 0.1$ and note that $0.1 \log |H| \leq 0.49 \log \frac{|H|}{2}$. If there exists a polynomial time $0.49 \log \frac{|H|}{2}$ approximation algorithm for the instance, then there exists a polynomial time $0.98 \log \frac{|H|}{2} \leq 0.98 \log |U|$ approximation algorithm for SETCOVERHARD, which is a contradiction to Lemma B.1. $\square$

