# OpenReview forum: "A Competitive Algorithm for Agnostic Active Learning"
_NeurIPS.cc/2023/Conference — NeurIPS 2023 poster_

### Official Review · Reviewer_orr4 · 2023-06-25

**Soundness:** 3 good
**Presentation:** 2 fair
**Contribution:** 1 poor
**Rating:** 7
**Confidence:** 3

**Summary:**

This paper studies agnostic active learning with a finite hypothesis space. The goal is to achieve a competitive query complexity with an optimal algorithm.

**Strengths:**

Interesting and important problem.

The approximation hardness based on Set-Cover is a nice addition.

--- rebuttal comment ---
After reviewers clarified concerns and added more results, I raised my score from 3 to 7.

**Weaknesses:**

The authors seem to be unaware of "Minimax analysis of active learning" by Steve Hanneke and Liu Yang [JMLR 2015]. Hanneke and Yang introduce the star number (a combinatorial dimension), which characterizes various active learning settings (realizable, different noise settings, including the one here) leading to near-tight lower and upper bounds. This seems to make the analyses here a special case of the achieved results. E.g., on a quick look Theorem 8 seems to have a similar competitive ratio as the proposed algorithms: roughly $d\log (1/\varepsilon)$ (with $d = VC$) instead of $\log |H| \geq d$.

Similarly, the discussed bounds $\mathcal{O}(m\log H)$ have been devised by Hegedűs, Tibor. "Generalized teaching dimensions and the query complexity of learning." [COLT 1995] long before Dasgupta and Nowak.

Claiming polynomial runtime is misleading, as one typically would mean polynomial in the instance space $X$ (or the sample, like in standard PAC) and not the hypothesis space $\mathcal{H}$, as is here the case.

**Questions:**

The authors write $|H|$ multiple times implying that the hypothesis space is finite but also use packings / coverings of $H$. Do the results generalize to hypothesis spaces with infinite size?

**Limitations:**

See weaknesses.

---

> ### Author Rebuttal · Authors · 2023-08-09
>
> **Related work.** We appreciate the additional references, which we will include in the
> paper and discuss.  But you are wrong about the relationship between our results and those of Hanneke-Yang '15. They actually give a much weaker
> form of ``optimality''.  See our general response for further
> discussion, and an example where their algorithm takes $H$ queries
> while ours takes $\log H$.
>
> **Runtime.** For the runtime, also see our general response.  We can make the
> theorem statement more explicit, if you feel it is misleading.
>
> **On $|H|$ vs a net, and whether H can be infinite in size.**
> The *core* of our algorithm expects a finite hypothesis space, and
> gives an $O(\log |H|)$ approximation.  But the algorithm does work on
> general hypothesis spaces with infinite size: the very first step of
> the algorithm is to find a $2\eta$-packing of $H$, and thereafter only
> consider this net.  So as long as you can compute a finite packing,
> the algorithm applies.

---

### Official Review · Reviewer_YT5S · 2023-06-28

**Soundness:** 3 good
**Presentation:** 2 fair
**Contribution:** 3 good
**Rating:** 7
**Confidence:** 3

**Summary:**

This paper applied a multiplicative weight update /  generalized binary search style algorithm to solve agnostic active learning. It proposes a novel "capping" approach to the weight (over hypotheses) to ensure the potential function always grows by some amount, and it proves this amount is lower bounded by $\Omega(1/m^*)$ where $m^*$ is the theoretical optimal instance-dependent label complexity. In the end, it shows that the proposed algorithm is competitive in the sense that its label complexity is at most, roughly speaking, $\tilde{O}(m^* \log |H|)$. And it also shows that removing the extra "log(|H|)" factor is NP-hard.

--

I've read the author's response. It addressed my major concern about related work so I increased my score to 7. I would encourage the authors to provide a comprehensive comparison with related works, including directly comparing bounds in different settings. I'm still not very convinced about the argument around computational efficiency, but that should not block this paper from being accepted.

**Strengths:**

- The proposed algorithm is a novel and nontrivial modification of the classical multiplicative weight update algorithm. The analysis from a competitive perspective also looks new to me in the setting of agnostic active learning.

- I briefly checked the proof and they look sound to me.

- Understanding the theoretical lower and upper bounds of instance dependent label complexity bounds for agnostic active learning is an interesting problem, and this paper makes a solid contribution by providing an algorithm whose label complexity is close to the optimal with an additional log(|H|) factor.

**Weaknesses:**

1. The additional factor of log(|H|) makes the label complexity of the proposed algorithm look not very strong to me: it can be worse than existing algorithms in many scenarios. The argument that getting rid of this factor is NP-hard does not look very relevant to me, since the main contribution of this paper is more on the information-theoretic side (the proposed algorithm is already by no means computationally efficient in the sense that it is polynomial in the number of hypotheses (or its epsilon cover at best) and the size of example space X), and it is known that even learning linear classifier without too much assumptions on noise is NP-hard.

2. It would clearer if there were more discussion about how good the label complexity is by comparing it with known upper and lower bounds in different settings. Especially, there are a few papers (e.g. Hanneke, Steve, and Liu Yang. "Minimax analysis of active learning." J. Mach. Learn. Res. 16.1 (2015): 3487-3602.) that give minimax analysis of active learning. I would like to see how your results compared with theirs. It would also be interesting if you could mention how the proposed algorithm works with noise assumptions (bounded, Tsybakov, etc.).

3. This paper focuses on the case where $\epsilon \gtrapprox \eta$, but it looks to me the case where $\epsilon$ is smaller can also be interesting (though of course the label complexity won't be exponentially better w.r.t. 1/\epsilon). It would be interesting if the theoretical results could cover that as well.

4. This paper is overall clear, but some wordings are a bit confusing to me unless I check the detailed proofs. E.g., the proof sketch in line 149 - 154 ("algorithm", "majority label", "inputs", "independent").

**Questions:**

I would like the authors to comment on the weaknesses 1~3 I mentioned above.

---

> ### Author Rebuttal · Authors · 2023-08-09
>
> **1. On the relevance of time complexity.** We respectfully disagree that our contribution is mainly on the
> information theoretic side.  We view it as a structural observation of
> how one can adapt a Bayes-inspired/multiplicative weights algorithm to
> get the frequentist agnostic learning guarantee.
>
> From an information theoretic perspective, one could do an exponential
> brute force like "consider all possible algorithms, and all possible
> settings of labels, and find the minimax optimal algorithm".  This
> would work, optimally, without saying anything about the problem.  The
> polynomial restriction on the runtime forces us to learn something
> about the problem.
>
> There are interesting cases (e.g. noisy binary search) where H is
> small enough that the algorithm is feasible.  And beyond that, we
> believe it should be feasible to adapt our algorithm to some
> structured infinite H; certainly multiplicative weights has been
> widely adapted.
>
> See our general response for more on why we think a polynomial
> dependence in this general problem is interesting.
>
>
> **2.  Related work.** See the general response for an example comparing our result to Hanneke-Yang '15.
>
> For weaker noise models, the optimal complexity could be much smaller,
> and our algorithm won't always match it.  For example, suppose there
> is $\eta/10$ probability mass on extremely informative points, with the
> rest of points being very uninformative.  In the agnostic setting, you
> can't trust the informative points because the adversary probably
> messes them all up, and so our algorithm won't sample them very often.
> In the bounded noise setting, you should just sample those points many
> times.
>
> **3.  Another regime.** Yes, $\varepsilon < \eta \ll 1$ could be interesting, where we suppose
> given existing bounds the goal would be to save a $\Theta(\eta)$
> factor relative to the passive ERM.
>
> It's a pretty different regime, though.  Consider the noisy binary
> search input (i.e., the hypothesis class is 1d threshold functions):
> for $\varepsilon > \eta$, you do some robust binary search to home in
> on the correct $O(\varepsilon)$ region in $O(\log \frac{1}{\eta})$
> queries.  For $\varepsilon < \eta$, you start with this phase to get
> within $O(\eta)$ of the truth, then do passive sampling over the
> $O(\eta)$-size region of ambiguity.  With $N$ samples you will
> typically get error $\frac{\eta}{\sqrt{N}}$ on each hypothesis, so you
> need about $O(\frac{\eta^2}{\varepsilon^2} + \log \frac{1}{\eta})$
> queries.
>
> So for noisy binary search there's two phases: one covered by this
> paper, and one that's no-longer-adaptive to refine $\varepsilon$.  It
> seems plausible that general problems can be solved with such an
> approach.

---

> > ### Comment · Reviewer_YT5S · 2023-08-15
> > **Thanks for the response!**
> >
> > I'm satisfied with the response about "Relation to Hanneke-Yang '15", but I would like to see your comment to the point about the extended teaching dimension raised by reviewer orr4.
> >
> > I have a few follow up questions:
> >
> > - Re:  "On the relevance of time complexity": It is still unclear to me why "an exponential time optimal algorithm would be trivial by brute force". In particular, we want an algorithm with low label complexity w.r.t. an **unknown** distribution, I don't see how brute force could deal with that. Is there any reference to this aspect?
> >
> > - Could you comment on my point that "the additional factor of log(|H|) makes the label complexity of the proposed algorithm look not very strong"?

---

> > > ### Author Response · Authors · 2023-08-15
> > >
> > > See our response to orr4 about the distributional extended teaching dimension result of Hanneke '07.
> > >
> > > Q1:  We want a low label complexity w.r.t. an unknown distribution over $Y \mid X$, relative to the optimal label complexity over all distributions over $Y \mid X$.  So for every algorithm, we can determine its performance on every distribution over $Y \mid X$, then take the algorithm with minimax complexity.  The running time is horrendous, but it's computable to arbitrarily good approximation.
> > >
> > > Q2: Our NP hardness result shows we could not hope for a polynomial time algorithm that avoids the $\log |H|$ factor.
> > > It's the best one can do even in the realizable, exact setting.
> > >
> > > Moreover:
> > >
> > >  1. Competing algorithms (e.g. by Hanneke) cannot even achieve that factor.
> > >  2. There are interesting cases, like noisy binary search, where it isn't too big.
> > >  3. There are cases (also like noisy binary search) where the algorithm outperforms the generic bound.

---

### Official Review · Reviewer_LeWc · 2023-07-07

**Soundness:** 4 excellent
**Presentation:** 4 excellent
**Contribution:** 3 good
**Rating:** 7
**Confidence:** 4

**Summary:**

The paper studies agnostic active learning by proposing a competitive algorithm that achieves at most a $\log H$ multiplicative factor on top of the optimal query lower bound of $m^*$. While similar result was known for the realizable setting, this paper makes a step toward understanding the agnostic setting. As a complementary result, it also shows that it is NP-hard to improve the $O(\log H)$ overhead both in realizable and agnostic settings.


**Strengths:**

The paper proposed a strategy that actively learns the target function in the agnostic setting with competitive sample complexity. Given that similar strategies only handle the realizable setting, it makes a significant contribution to the active learning problems in general settings. The proofs seem sound. The paper is well written.


**Weaknesses:**

The paper is clear in the description of the algorithm and the analysis of how it queries. However, the computational cost of the algorithm is vague. In Theorem 1.1, it claims that the algorithm runs in polynomial time, but it seems not so at least in the dimension parameter d.


**Questions:**

Can the author comment on the exact running time of the algorithm on each parameter, for example, $d, m, |H|, |X|, \frac{1}{\epsilon}$, (or any other important parameters)?


**Limitations:**

No concerns.

---

> ### Author Rebuttal · Authors · 2023-08-09
>
> **Runtime.** We should give the runtime more precisely, and appreciate the comment.  The main cost is
> from checking if any $\epsilon$-ball of hypotheses has 80\%
> probability, after each label seen.  Naively this takes about
> $O(|H|^2(|\mathcal{X}| + m))$ time, because it takes $|H|^2|\mathcal{X}|$ time to compute the
> distances between all hypotheses and then $|H|^2 m$ time to try
> all the balls in all iterations.
>
> However, one should be able to optimize this by making that step
> randomized and approximate.  This would make the analysis more
> annoying, but shouldn't change the result other than the constant
> factors.  One could just sample $O(\log \frac{1}{\delta'})$ random
> hypotheses, and $O(1/\epsilon)$ random $x$, and see if at least 90\%
> of the hypothesis pairs are empirically within $O(\epsilon)$ of each
> other.
>
> With this optimization, the algorithm should have an overall runtime
> around $\tilde{O}(|H| m + m / \epsilon)$: the first term is
> for $m$ rounds of multiplicative weights over $|H|$, and the second term is
> the optimized identification of heavy balls. Plus, if you want a net to improve the approximation from $O(\log |H|)$ to $O(\log |N(H, \eta)|)$, however much time it takes to compute this net at the beginning (at most $|H|^2/\eta$ with a greedy algorithm, but typically you would write it down explicitly for your hypothesis class).
>
> But the analysis of this variant is a little hairy, and in our opinion
> optimizing the algorithm for generic hypothesis classes is not as interesting as getting much more significant speedups
> for specific hypothesis classes.

---

> > ### Comment · Reviewer_LeWc · 2023-08-20
> > **Official Comment by Reviewer LeWc**
> >
> > Thank you very much for your response. It is very clear to me now. I encourage the authors to add these comments in their revision. My review and score remain unchanged.

---

### Official Review · Reviewer_wAoo · 2023-07-10

**Soundness:** 4 excellent
**Presentation:** 2 fair
**Contribution:** 2 fair
**Rating:** 4
**Confidence:** 3

**Summary:**

The authors provide an algorithm for learning in the presence of agnostic noise. The algorithm finds a hypothesis that gets error $O(\eta)$ where $\eta$ is the noise parameter. The algorithm requires querying specific points, so it uses a stronger oracle than the standard active learning. The algorithm is possibly inefficient for many classes as it is polynomial in the number of hypotheses (or to the size of the cover). Their algorithm is optimal up to log factors, and they provide a lower bound showing that it is NP-Hard to improve this.

**Strengths:**

This work provides a query-efficient algorithm with tight results (upper and lower bound) for the agnostic setting.

**Weaknesses:**

The setting in their analysis is not active learning but learning with queries which is a stronger oracle. So the title of the paper/abstract is misleading. In my opinion, active learning is a completely different model than the one the authors use. Also for examples the 41-46 lines, the papers there are for active learning and not for the model the authors analyze. I think the authors should be more clear about this. About the NP-Hardness result. This result makes sense when the $|H|$ is polynomial over $\epsilon$ and the dimension. But almost always the $|H|$ is exponentially large, i.e., $d$-dimensional functions.


**Questions:**

N/A

**Limitations:**

No liomitations.

---

> ### Author Rebuttal · Authors · 2023-08-09
>
> Definition of Active Learning
> -------
>
> You appear to be concerned that we assume we know $D_X$ and can query
> $(Y | X = x)$, while some previous work defines active learning as:
> given all the $x_i$ in a dataset, pick a subset of them to see the
> corresponding $y_i$.  However, in the limit of infinitely many
> unlabeled data points, these are equivalent: the set of all $x_i$
> gives us the distribution $D_X$, and we can sample any $x$ in our
> dataset, which means any $x$ in the support. So if we don't care about
> the number of unlabeled examples, as the prior work also doesn't, then
> we may as well assume our model.
>
> Bounds on the unlabeled data complexity are a very interesting
> question which we intend to pursue in followup work.  It gets somewhat
> complicated, because the OPT also depends on the amount of unlabeled
> data, so we believe focusing on labeled complexity makes sense.
>
>
>
> About the NP-Hardness result
> --------
>
> As we discuss in the general response, at our level of generality one
> cannot avoid a runtime polynomial in $|H|$.  And without the
> $\log |H|$ slack in number of samples, our NP-hardness result shows one
> cannot hope to avoid time *exponential* in $|H|$.
>
> Faster algorithms under structural assumptions on $H$ are a good
> direction for future work, but our results here are the best one could
> hope for.

---

### Author Rebuttal · Authors · 2023-08-09

We thank the reviewers for their comments.  We would like to emphasize that we give the first algorithm for active agnostic learning that is competitive with the optimal algorithm for a given input (unlabeled data and hypothesis class).  There are a couple general points we would like to clarify, particularly about a prior minimax work by Hanneke and Yang which does *not* get this.

Relation to Hanneke-Yang '15
-------------------------

A couple reviewers pointed to the minimax analysis of Hanneke-Yang
'15, which we had missed.  Thanks!  The distinction is that the
Hanneke-Yang upper bound is close to the optimal complexity of the
*worst case $D_X$*, while ours is relative to the optimal complexity
of our actual $D_X$.  As a result, our algorithm can be *much* better,
as the following example shows:


Define a hypothesis class of $N$ hypotheses $h_1, \dotsc, h_N$, and $(N + \lg
N)$ data points $x_1, \dotsc, x_{N + \lg N}$, so for each hypothesis $h_j$:

 * The labels of the first $N$ points express $j$ in unary
 * The labels of the last $(\lg N)$ points express $j$ in binary

So, for example, when N = 16 then $h_6$ will have labels:

  00000100000000000110

  Now consider the realizable ($\eta = 0$) setting, and suppose that
  the $x_i$ are fairly uniform (e.g., min probability
  $\frac{1}{10 N}$) and $\epsilon = \frac{1}{20 N}$.  Then you need to
  identify $h_j$ exactly by querying the labels.

The obvious, optimal strategy is: query the last $\lg N$ bits and read
off the answer.  Our algorithm will basically do this, getting
$\Theta(\lg N)$ complexity.  RobustCAL, the algorithm analyzed in HY15,
will not: the first $O(N/\lg N)$ points it sees will be in the unary
region, and it will label almost every one of them, for $\Theta(N/\lg N)$
complexity.


So in this example, our algorithm is optimal at $O(\lg N)$, and our
general theorem's guarantee is $O(\lg^2 N)$ which is pretty close.
HY15's algorithm actually takes $\Theta(N/\lg N)$ queries, and their
general guarantee is $O(N \lg^2 N)$.


You might be confused: how can HY15 claim minimax optimality for an
algorithm with such poor behavior?  Their minimax lower bound on this
example is actually $N$.  This is because if $D_X$ happened to be
uniform on the first $N$ points (i.e., only the unary ones), you
really would need $N$ queries to recover the hypothesis.

But in active learning, you *know* your unlabeled data $x_i$.  So the
algorithm can see that it has access to the great points at the end
and should choose to query them.  Our algorithm does; RobustCAL and $A^2$
do not.


On Polynomial Running times
-----


A couple reviewers point out that usually $|H|$ is very large, so a
polynomial dependence on $|H|$ isn't great.  However:

* The problem setting we consider is fully general hypothesis classes.
Therefore the *input* has size linear in $|H|$, so one cannot hope to
avoid this dependence.

* There are interesting cases (e.g. binary search) where the
hypothesis class isn't huge, so the algorithm is feasible.

* The prior work in this setting (e.g. $A^2$) also depends polynomially
on $|H|$.

* Getting an *exponential time* optimal algorithm would be trivial by
brute force, so requiring polynomial time is what forces us to
understand something interesting about the problem.


We certainly agree that getting faster algorithms for more restrictive
hypothesis classes is an important line of future work.

---

> ### Comment · Reviewer_orr4 · 2023-08-14
>
> Good points, you are right, Hanneke and Yang's star number and disagreement coefficient in general do not seem to yield the desired results, since they take a worst-case distribution (and potentially sample).
>
> However, your given example is in the finite and realizable case. In such a case the extended teaching dimension $\text{EXT-TD}(H)$ guarantees near-tight sample-dependent bounds. $\text{EXT-TD}(H)$ is a lower bound and $O(\text{EXT-TD}\log|H|)$ as an upper bound, see Hegedűs [1995], exactly the $O(\log|H|)$ guarantee you state.
>
> Subsequently, Hanneke achieved similar distribution dependent bounds by adapting the extended teaching dimension to the agnostic case. E.g., Theorem 4 in Hanneke [2007] seems to yield a similar results to you.
>
> Please clarify, if there is some misunderstanding. I just want to understand to what extend the achieved results are novel.
>
> Another quick question: Your main algorithm seem to require to know $m^*$ (the optimal number of samples), which is of course not true in general. Your bounds are only interesting if $m^*$ is unknown. But I guess you can apply some sort of doubling trick?
>
> Thanks!
>
> Best,
>
> orr4
>
> Hegedűs, Tibor. "Generalized teaching dimensions and the query complexity of learning" COLT 1995
>
> Hanneke, Steve "Teaching Dimension and the Complexity of Active Learning" COLT 2007

---

> > ### Author Response · Authors · 2023-08-15
> >
> > For sure, the example above is in the realizable, exact case which has been solved for decades.  The point of our paper is to get similar bounds for the agnostic case.  But the prior work for the agnostic case can't even solve the example above.
> >
> > Hanneke '07
> > -----
> >
> > Hanneke '07 fails both algorithmically and analytically on the same example, for $\eta = \epsilon = 3/N$.
> >
> > You point specifically to Theorem 4 of Hanneke '07, but this has exactly the same issue as Hanneke-Yang '15:  the lower bound is for the worst-case distribution over $X$.  The relevant line of Theorem 4 of Hanneke '07 is:
> > $$
> > XPTD(V,(1-\epsilon)/\epsilon, \delta) \leq \sup_{D' \in \mathcal{D}} LQ(\mathbb{C}, D', \epsilon, \delta, 0)
> > $$
> > which involves a supremum over all distributions $D'$ supported on $\mathcal{X}$.
> >
> > So there *exists* a distribution over the support such that the upper and lower bounds are both related to teaching dimension, but it probably isn't the given distribution.   (The bound is also worse in that the lower bound is *partial* teaching dimension, which can be much smaller, particularly if $\delta$ is reasonably large.)
> >
> > And it's not just an analytical issue, if you read through the Algorithm ReduceAndLabel in Hanneke '07.   Consider the same example as above, but with $\eta = \epsilon = \frac{3}{N}$.  The optimal algorithm is still $O(\log N)$ queries: just query the last $\lg N$ bits and output that hypothesis, which might be slightly wrong in the unary region but has error $\eta + \frac{2}{N}$.
> >
> > In Hanneke '07, the first step is to take a sample of $\frac{X}{5 \log |H|}$ examples; in the example above, this is probably going to be entirely in the unary region.  Then the algorithm calls Reduce on this sample, which repeatedly takes $O(1/\eta)$-size subsamples and labels their minimal specifying set; but if the original sample is unary, this set will be of size $O(1/\eta)$.  So Hanneke '07 will use a linear number of queries on this example.
> >
> > Wait, you might say, what about Theorem 3 of Hanneke '07?  That's an upper bound in terms of extended teaching dimension that doesn't involve a supremum over distributions.  But it's analyzing the algorithm that we know fails on this example, so of course it has to be a weak upper bound.  Why?
> >
> > The issue is that the distributional version of extended teaching dimension considered in Hanneke '07 isn't terribly good.  It's in terms of a parameter $n$, which is set to about $\frac{1}{16 \eta}$; the distributional extended teaching dimension basically says: if you only had $n$ unlabeled data points from $\mathcal{D}$, how many labels would you need to teach?  And in the example, with only $O(N)$ unlabeled data points you will probably still be missing a constant fraction of the binary points, so the teaching dimension remains polynomially high.  If you have access to $\gg \frac{1}{\eta}$ unlabeled samples, you can hope to do much better.
> >
> > [*] It's a minor point, but I think there's an error in the proof of Theorem 3 in Hanneke '07 that slightly affects exactly what polynomial dependence it gets on this example.  I think the actual algorithm takes $\tilde{\Omega}(N)$ queries, but the theorem statement would give an $N^{0.99}$ upper bound because you would see a constant fraction of those binary points.  The algorithm samples $U_i$ from $U$, then subsamples that *with replacement* and finds a minimal specifying set.  The distributional extended teaching dimension bounds the size of the specifying set whp for a *single round* of subsampling, but the double-round subsampling is not exactly the same distribution.  The proof of Lemma 3 doesn't do anything to address this issue (it just says "For each, we make at most $t$ label requests with probability $\geq 1 - \frac{\delta}{2s}$", which would be true by assumption for the single round sampling), and in fact on the example with $\eta = \epsilon = 3/N$ it matters a bit.  But regardless of whether the bound is $N$ or $N^{0.99}$, it's not the logarithmic dependence of optimal or our algorithm.
> >
> >
> > On the need for knowing $m^*$
> > -----
> >
> > You're right that our algorithm is written to assume $m^\*$, but it's a pretty minor dependence that can be removed.  Thanks for pointing it out.
> >
> > The algorithm only uses $m^\*$ to set the number of iterations $k$ in the first stage.  One could modify the algorithm to remove this dependence, as follows:  $m^\*$ enters the proof through Lemma 2.1, which combined with (1) gives a lower bound on the expected potential gain in each round.  But the quantity being bounded (LHS of Lemma 2.1 / RHS of (1)) does not depend on $h^\*$ or $m^\*$, and can be computed by the algorithm.  So the algorithm *knows* a lower bound $\tau_i \gtrsim \frac{1}{m^\*}$ on how much the potential is expected to grow in each round $i$.  Then the algorithm can stop as soon as $\sum_{i=1}^k \tau_i \geq O(\log H)$, at which point the potential is positive whp and the second stage will succeed.

---

> > > ### Comment · Reviewer_orr4 · 2023-08-17
> > >
> > > Thank you for the clarifications. I now think that this paper is valuable to the active learning community at NeurIPS and hence changed my vote to accept.

---

### Decision · Program_Chairs · 2023-09-21

**Decision:**

Accept (poster)

**Comment:**

This paper provides a general algorithm and approximation ratio analysis for the setting of agnostic active learning with an arbitrary hypothesis class. This is a significant contribution to the active learning field, though providing a simpler and more computationally efficient algorithm (perhaps some sort of oracle access to H that can be done in O(|H|) time) and an approximation ratio for the same \eta and \epsilon would be even more significant.